

# Multi-objective optimization for smart cities: a systematic review of algorithms, challenges, and future directions

YiFan Chen[1,2], Weng Howe Chan[2,3], Eileen Lee Ming Su[4] and Qi Diao[5]

[1] Jiaxing Key Laboratory of Industrial Intelligence and Digital Twin, Jiaxing Vocational and Technical College, Jiaxing, Zhejiang, China
[2] Faculty of Computing, Universiti Teknologi Malaysia, Johor Bahru, Johor, Malaysia
[3] UTM Big Data Centre, Ibnu Sina Institute for Scientific and Industrial Research, Universiti Teknologi Malaysia, Johor Bahru, Johor, Malaysia
[4] Faculty of Electrical Engineering, Universiti Teknologi Malaysia, Johor Bahru, Johor, Malaysia
[5] Faculty of Artificial Intelligence, Zhejiang Dongfang Polytechnic, Wenzhou, China

Corresponding authors
YiFan Chen, chenyifan@jxvtc.edu.cn
Weng Howe Chan, cwenghowe@utm.my

## ABSTRACT

With the growing complexity and interdependence of urban systems, multi-objective optimization (MOO) has become a critical tool for smart-city planning, sustainability, and real-time decision-making. This article presents a systematic literature review (SLR) of 117 peer-reviewed studies published between 2015 and 2025, assessing the evolution, classification, and performance of MOO techniques in smart-city contexts. Existing algorithms are organised into four families—bio-inspired, mathematical theory-driven, physics-inspired, and machine-learning-enhanced—and benchmarked for computational efficiency, scalability, and scenario suitability across six urban domains: infrastructure, energy, transportation, Internet of Things (IoT)/cloud systems, agriculture, and water management. While established methods such as Non-dominated Sorting Genetic Algorithm II (NSGA-II) and Multiobjective Evolutionary Algorithm based on Decomposition (MOED/D) remain prevalent, hybrid frameworks that couple deep learning with evolutionary search display superior adaptability in high-dimensional, dynamic environments. Persistent challenges include limited cross-domain generalisability, inadequate uncertainty handling, and low interpretability of artificial intelligence (AI)-assisted models. Twelve research gaps are synthesised—from privacy-preserving optimisation and sustainable trade-off resolution to integration with digital twins, large language models, and neuromorphic computing—and a roadmap towards scalable, interpretable, and resilient optimisation frameworks is outlined. Finally, a ready-to-use benchmarking toolkit and a deployment-oriented algorithm-selection matrix are provided to guide researchers, engineers, and policy-makers in real-world smart-city applications. This review targets interdisciplinary researchers, optimisation developers, and smart-city practitioners seeking to apply or advance MOO techniques in complex urban systems.

# INTRODUCTION

With the rapid advancement of global urbanization, smart cities have emerged as a core strategy to promote sustainable urban development, attracting a large amount of attention from academia and industry. By integrating cutting-edge technologies such as the Internet of Things (IoT), big data analytics, and artificial intelligence (*Hashem et al., 2016*; *Allam & Dhunny, 2019*; *Bibri & Krogstie, 2019*; *Yigitcanlar et al., 2023*), smart cities aim to achieve efficient urban operations and optimal resource utilization, thus improving the quality of life of residents and fostering environmental sustainability (*Dhiman & Alghamdi, 2024*).

Smart cities integrate subsystems like energy, transportation, environment, and infrastructure to optimize urban efficiency and sustainability. These subsystems are interdependent, and decisions in one area often affect performance in others (*Gallotti, Sacco & De Domenico, 2021*; *Branny et al., 2022*; *Ammara et al., 2022*; *Bibri, Huang & Krogstie, 2024*). Achieving goals such as improving energy efficiency, optimizing traffic flow, and enhancing environmental protection is critical for smart city development. However, these objectives often conflict during implementation, requiring careful trade-offs among resource utilization, operational costs, and environmental sustainability (*Syed et al., 2022*; *Sawik, 2023*; *Tang & Osaragi, 2024*). Addressing these trade-offs effectively has become a key issue in smart city research and practice.

The integration of multi-objective optimization (MOO), hereafter referred to simply as MOO, in smart cities has already led to measurable improvements in key urban sectors worldwide. For instance, in Tokyo, Japan, MOO-driven evacuation planning has increased efficiency by 20% while minimizing fire hazard risks in post-earthquake scenarios (*Tang & Osaragi, 2024*). In Beijing, an optimization framework for urban planning has led to enhanced thermal comfort and a reduction in urban heat island effects, balancing environmental sustainability with infrastructural needs (*Xiao, 2024*). Similarly, in Zurich, participatory MOO models for urban land use planning have optimized dense and green city layouts, enhancing both spatial efficiency and ecosystem service preservation (*Wicki et al., 2021*).

As computational power, AI-driven heuristics, and real-time data integration continue to advance, MOO is evolving from single-domain optimizations to holistic, cross-sectoral coordination models. Future smart cities will leverage quantum computing, deep reinforcement learning, and large-scale data-driven decision-making to achieve unprecedented levels of urban resource management and policy optimization. This paradigm shift offers policymakers an intelligent, adaptive framework to dynamically balance sustainability, efficiency, and economic growth, marking a revolutionary step in urban governance and decision-making (*Masoumi & van Genderen, 2024*).

MOO provides an effective framework for resolving these conflicts by simultaneously optimizing multiple objectives (*Othman, Darwish & Abd El-Moghith, 2023*; *Li et al., 2023*). Through the generation of Pareto-optimal solutions, MOO allows decision-makers to evaluate trade-offs and select solutions that align with specific contextual priorities (*Zaizi, Qassimi & Rakrak, 2023*; *Masoumi & van Genderen, 2024*; *Xie et al., 2024*).

## Motivation and main contributions

Smart cities face increasingly complex decision-making challenges that involve balancing multiple conflicting objectives—such as efficiency, sustainability, resilience, and inclusiveness. Despite a growing body of algorithmic developments in MOO, there remains a lack of systematic synthesis across urban domains. This study is motivated by the need to consolidate fragmented research and provide a structured landscape for future developments. MOO techniques have significantly advanced smart city management by optimizing subsystems, yet addressing the complex, dynamic, and interdependent nature of urban environments remains a major challenge. Smart cities operate within large-scale, high-dimensional datasets and continuously evolving operational demands, requiring optimization techniques that balance computational efficiency, real-time adaptability, and multi-criteria trade-offs. Although state-of-the-art MOO algorithms such as Non-dominated Sorting Genetic Algorithm II (NSGA-II), Multiobjective Evolutionary Algorithm based on Decomposition (MOED/D), and particle swarm optimization (PSO) have shown success in specific smart city applications, their performance often declines in highly dynamic and large-scale real-time scenarios.

Despite growing research interest, existing studies predominantly focus on isolated applications or single algorithmic approaches, leading to a fragmented understanding of MOO in smart cities. This lack of a unified perspective hinders algorithm selection for urban planners and decision-makers seeking the most suitable optimization approach for a given scenario. Additionally, the absence of standardized performance metrics and benchmarking frameworks makes it difficult to compare the computational efficiency, scalability, and real-world applicability of different MOO techniques. These challenges highlight the pressing need for a comprehensive and systematic analysis that not only categorizes MOO algorithms but also evaluates their performance across diverse smart city applications under a unified framework.

To address these gaps, this study undertakes a systematic literature review (SLR), hereafter referred to simply as SLR, focusing on the classification, evaluation, and applicability of MOO algorithms across diverse smart city domains. The core contributions of this study are summarized as follows:

1. **Unified taxonomy of 117 MOO algorithms.** The review organises 117 algorithms into four principal families—bio-inspired, mathematical theory-driven, physics-inspired, and machine-learning-enhanced—creating a common reference frame for subsequent performance and applicability analyses.

2. **Scenario-based comparative assessment.** Six core smart-city domains (infrastructure, energy, transportation, IoT/cloud, agriculture, and water) are benchmark-evaluated with respect to convergence, diversity, and computational complexity, revealing domain-specific strengths and trade-offs of each algorithm family.

3. **Standardised benchmarking toolkit.** The study consolidates public datasets, objective dimensionalities, and Pareto indicators, providing a ready-to-use toolkit that supports reproducible and cross-domain algorithm evaluation.

4. **Gap analysis and research roadmap.** Twelve unresolved challenges—covering generalisation, real-time responsiveness, interpretability, uncertainty, and privacy—are synthesised, and targeted research avenues such as privacy-aware optimization, sustainability-centric models, and hybrid MOO-LLM frameworks are proposed.

5. **Deployment-oriented decision guidance.** By mapping algorithm traits to specific urban requirements, the review offers a structured selection matrix that aids researchers, engineers, and policymakers in choosing fit-for-purpose MOO techniques for real-world smart-city deployments.

The remainder of the article is organized as follows: "Survey Methodology" presents the methodology, including the SLR protocol and selection criteria. "Findings of the SLR" outlines the major findings, covering algorithm classification, dimensional objective conflicts, and performance comparisons. "Challenges and Gaps in Current MOO Research" discusses key challenges and existing research gaps, followed by "Future Perspectives and Needs", which proposes a structured agenda for future work. "Conclusion" concludes the article with a summary of contributions and implications for multi-objective optimization in smart cities.

This review is intended for a multidisciplinary audience, including researchers, engineers, and decision-makers involved in urban computing, optimization methodologies, and sustainable infrastructure design. The study aims to deliver both theoretical insight and practical guidance for effective MOO integration in smart urban systems.

## SURVEY METHODOLOGY

This study applies the SLR approach to identify key research gaps in MOO applications in smart cities. With numerous studies published annually and often presenting conflicting conclusions, SLR synthesizes findings to advance understanding by resolving inconsistencies (*Moher et al., 2009*; *PRISMA-P Group et al., 2015*). The methodology follows three core principles: (a) explicit methodology for transparency, (b) replicability for reproducibility, and (c) evidence synthesis to integrate diverse findings into a unified framework (*Cruz et al., 2024*). Through a transparent and structured framework, SLR fosters theory development, addresses research saturation, and highlights unexplored opportunities (*Hulland & Houston, 2020*; *Paul & Criado, 2020*). This review also adopts principles from literature-based discovery workflows to enhance methodological rigor and uncover hidden patterns (*Thilakaratne, Falkner & Atapattu, 2019*). Its strength lies in minimizing selection bias by including only studies that meet objective criteria (*Kitchenham et al., 2009*; *Pati & Lorusso, 2018*). Although SLR requires more time and effort than traditional reviews and is constrained by predefined keywords and search strings, it remains widely adopted for its methodological rigor, objectivity, and suitability for complex topics (*Paul et al., 2021*).

Following established methodologies, this SLR is structured into four stages: (1) Formulating research questions, (2) locating scientific studies, (3) selecting and evaluating relevant articles, and (4) synthesizing and reporting results. These stages, while

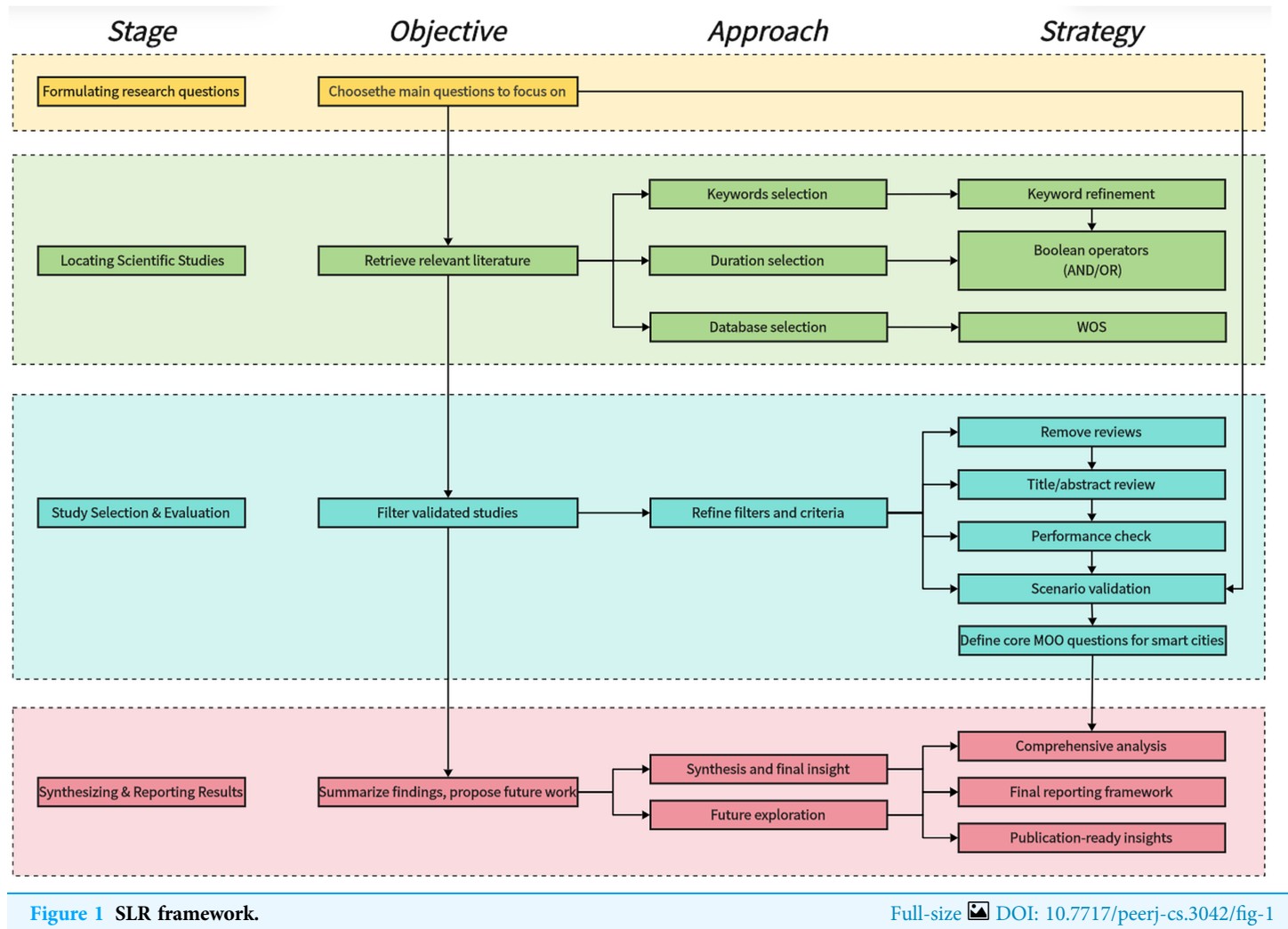

**Figure 1  SLR framework.**

presented sequentially, are inherently iterative insights from later steps often inform earlier ones. Figure 1 illustrates this process, summarizing the key strategies to ensure clarity, transparency, and methodological rigor.

## Formulating research questions

To systematically investigate the evolution, performance, and limitations of MOO in smart city contexts, this review is guided by three core research questions. These questions are designed not only to organize existing knowledge but also to synthesize cross-domain insights, reveal underexplored research opportunities, and establish a unified understanding of how MOO techniques are evolving to meet the demands of smart urban systems.

**RQ1: How are MOO algorithms categorized, and how have they evolved in response to smart city complexity?** This question explores the historical and methodological development of MOO algorithms, with a focus on algorithmic classification, performance

evolution across decades, and integration with emerging paradigms such as hybrid models, ML-assisted search, and real-time decision systems.

**RQ2: How do MOO algorithms perform across different smart city domains, and which strategies are most effective in resolving low-, mid-, and high-dimensional objective conflicts?** This question investigates the domain-specific applicability of MOO techniques, aiming to reveal the strengths and limitations of algorithm families under varying smart city conditions by analyzing their ability to resolve diverse levels of trade-off complexity.

**RQ3: What are the key limitations and research gaps in existing MOO applications, and how can future studies enhance scalability, adaptability, and ethical robustness?** This question aims to uncover both technical and ethical limitations in current MOO applications and to propose future directions for building scalable, generalizable, and interpretable optimization frameworks that are aligned with real-world urban needs.

## Locating scientific studies

In the second stage, the process of locating scientific studies began by defining keywords derived from core research objectives using a structured word hierarchy. An efficient translation of research questions into precise search terms and Boolean strings was conducted to maximize the retrieval of relevant articles (*Kitchenham & Brereton, 2013*; *Boell & Cecez-Kecmanovic, 2015*; *Petersen, Vakkalanka & Kuzniarz, 2015*).

This approach aligns with evidence-based recommendations for balancing recall and precision in systematic reviews (*Gusenbauer & Haddaway, 2020*). Synonymous and related terms were also considered to ensure comprehensive coverage of relevant research topics. Tables 1 and 2 outlines the complete search string employed in this SLR, ensuring transparency and reproducibility.

At this stage, individual or pairwise searches for specific smart city scenarios were deliberately excluded. This decision was justified by the substantial body of existing research in relevant areas and the overarching objective of identifying potential synergies among diverse optimization applications. Additionally, the search process was conducted using the Web of Science (WoS) database since it is widely regarded as the gold standard for citation analysis, particularly suitable for robust bibliometric research (*Harzing & Alakangas, 2016*; *Singh et al., 2021*). The final search was completed in February 2025, ensuring inclusion of the most recent peer-reviewed studies. Focusing on high-impact, peer-reviewed journal articles. The 2015–2025 timeframe aligns with the convergence of smart city technological maturity (*Bibri & Krogstie, 2017*), advancements in MOO algorithms (*Deb & Jain, 2014*; *Alam et al., 2020*), and global policy mandates prioritizing sustainable urban development (*European Commission, 2016*; *The State Council of China, 2021*).

## Study selection and evaluation

The third stage focused on refining the literature pool retrieved from the WoS database through the application of predefined inclusion and exclusion criteria (*Petersen, Vakkalanka & Kuzniarz, 2015*). Only peer-reviewed journal articles published between

**Table 1  Search strategy.**

| Category | Search terms | Boolean logic |
|---|---|---|
| MOO algorithms | TS = ("Multi-Objective Optimization" OR "MOO" OR "NSGA-II" OR "MOEA/D") | OR |
| Smart cities | TS = ("Smart Cities" OR "Urban Systems" OR "Smart Communities" OR "Sustainable Urbanization") | OR |
| Time filter | PY = (2015–2025) | — |

**Table 2  Final search string used for literature query.**

| Final search string |
|---|
| ("Multi-Objective Optimization" OR "MOO" OR "NSGA-II" OR "MOEA/D") |
| ("Smart Cities" OR "Urban Systems" OR "Urban Development" OR "Smart Communities" OR "Sustainable Urbanization") |
| (2015–2025) |

2015 and 2025 were considered to ensure methodological rigor and relevance to recent advancements in MOO. Articles were selected based on their alignment with the research questions outlined in the section "Formulating Research Questions", and their focus on smart city domains such as urban development, infrastructure, energy systems, transportation, and IoT. A two-step screening process-initial review of titles and abstracts followed by full-text evaluation of methodology and findings was used to identify studies that offered substantive contributions to MOO applications in smart cities (*Page et al., 2021*).

## Synthesizing and reporting results

A total of 117 articles were selected for in-depth analysis. These studies were thematically categorized by application scenarios, including energy management, urban infrastructure, transportation systems, IoT platforms, and environmental planning (*Batty et al., 2012*; *Bibri & Krogstie, 2017*; *Xiao, 2024*). This categorization enabled the identification of cross-cutting optimization goals, commonly adopted algorithms, and frequently used performance metrics (*Braun & Clarke, 2006*). The synthesis revealed several recurring gaps: insufficient integration of real-time data, lack of standardized performance benchmarks, and limited research on cross-scenario optimization frameworks (*Massoud Amin, 2011*; *Salimi et al., 2019*; *Jafari et al., 2023*). These insights lay the foundation for the subsequent discussion on the mathematical modeling and algorithmic classification of MOO techniques in smart city contexts.

## Theoretical background
### Fundamental model of MOO problems

Modern smart cities, as large-scale complex systems, inherently pose scientific challenges related to multi-objective coordinated optimization. MOO offers a rigorous computational approach to address key urban challenges, such as resource allocation, service scheduling, and operational efficiency by using mathematical models and algorithmic strategies. A

typical smart city optimization problem can be formalized as a vectorized multi-objective model:

$$
\begin{aligned}
\text{Minimize}: \quad & F(x) = [f_1(x), f_2(x), \ldots, f_m(x)]^T \\
\text{Subject to}: \quad & g_i(x) \leq 0, \quad i = 1, 2, \ldots, p \\
& h_j(x) = 0, \quad j = 1, 2, \ldots, q \\
& x \in \Omega
\end{aligned}
\tag{1}
$$

where the decision variable vector $x = [x_1, x_2, \ldots, x_n]^T$ represents controllable urban parameters, such as traffic signal timing, energy network topology, and infrastructure layout. The objective function vector is defined as $F(x) = [f_1(x), f_2(x), \ldots, f_m(x)]^T$, where each component $f_i(x)$ denotes a specific urban performance indicator, including traffic efficiency, carbon emissions intensity, and public service quality. In most urban systems, optimizing one performance metric often adversely affects others. Figure 2 presents two sample objective functions. As shown, improving one function often leads to degradation in the other, thus motivating the need for multi-objective trade-off analysis.

The optimization problem is further subject to a set of constraints. The inequality constraints $g_i(x) \leq 0$ $(i = 1, \ldots, p)$ and equality constraints $h_j(x) = 0$ $(j = 1, \ldots, q)$ represent physical limitations and operational rules embedded in urban systems.

For example, in urban logistics optimization, a common constraint is the vehicle capacity limit, which can be mathematically expressed as:

$$
\sum_{i=1}^{n} d_i x_{ik} \leq Q_k, \quad \forall k \in \{1, \ldots, K\}
\tag{2}
$$

where:

- $x_{ik} \in \{0, 1\}$ is a binary variable indicating whether customer $i$ is served by vehicle $k$;
- $d_i$ is the demand of customer $i$ (*e.g.*, in kg/m$^3$);
- $Q_k$ is the maximum load capacity of vehicle $k$;
- $K$ is the total number of available vehicles.

Due to the conflicting nature of multiple objectives, conventional single-objective optimization methods are insufficient. Instead, the concept of a Pareto-optimal solution set is used, comprising solutions in which no single objective can be improved without compromising at least one other.

### Pareto optimality and the Pareto front

In MOO, a solution is considered Pareto-optimal if it is not dominated by any other feasible solution in the search space. Formally, given two solutions $x^a$ and $x^b$, $x^a$ dominates $x^b$ (denoted $x^a \prec x^b$) if and only if:

$$
\forall j \in \{1, 2, \ldots, m\}, f_j(x^a) \leq f_j(x^b) \quad \text{and} \quad \exists k, f_k(x^a) < f_k(x^b).
\tag{3}
$$

This condition ensures that $x^a$ is no worse than $x^b$ across all objectives and strictly better in at least one. The set of all such non-dominated solutions forms what is known as the Pareto-optimal solution set.
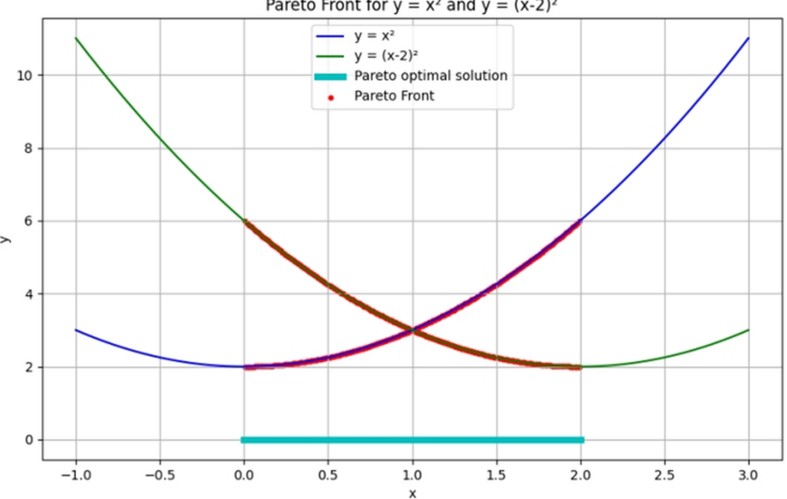

**Figure 2 Example of Pareto front for a quadratic MOO problem.**

To concretely illustrate this concept, consider the following bi-objective minimization problem:

$$f_1(x) = x^2 + 2, \quad f_2(x) = (x - 2)^2 + 2. \tag{4}$$

Within the interval $x \in [0, 2]$, no solution minimizes both objectives simultaneously. Instead, each solution represents a distinct trade-off between $f_1$ and $f_2$. The set of non-dominated solutions in this domain constitutes the Pareto-optimal set in the decision space.

Figure 3 illustrates the corresponding Pareto front in the objective space. Notably, at $x = 1$, both objectives achieve balanced values, *i.e.*, $f_1(1) = f_2(1) = 3$, demonstrating a characteristic equilibrium point.

Mathematically, the Pareto front is defined as the image of the Pareto-optimal decision set $\Omega^*$ mapped into the objective space by the vector-valued function $F(x)$:

$$\mathcal{F}^* = \{(f_1(x), f_2(x)) \mid x \in \Omega^*\}. \tag{5}$$

This front encapsulates all trade-off solutions that cannot be improved in one objective without compromising another. Consequently, identifying the Pareto front is a central goal of most MOO algorithms, forming the foundation for informed decision-making in real-world applications. However, in practice, the shape of the Pareto front can exhibit complex geometries, such as discontinuities, multimodality, or loss of dominance pressure, especially in high-dimensional urban contexts. These complex Pareto front geometries, such as discontinuities or weak dominance pressure, pose serious challenges to algorithm convergence and diversity preservation. Consequently, understanding these structures is essential for the design and evaluation of MOO algorithms, as further discussed in later section.

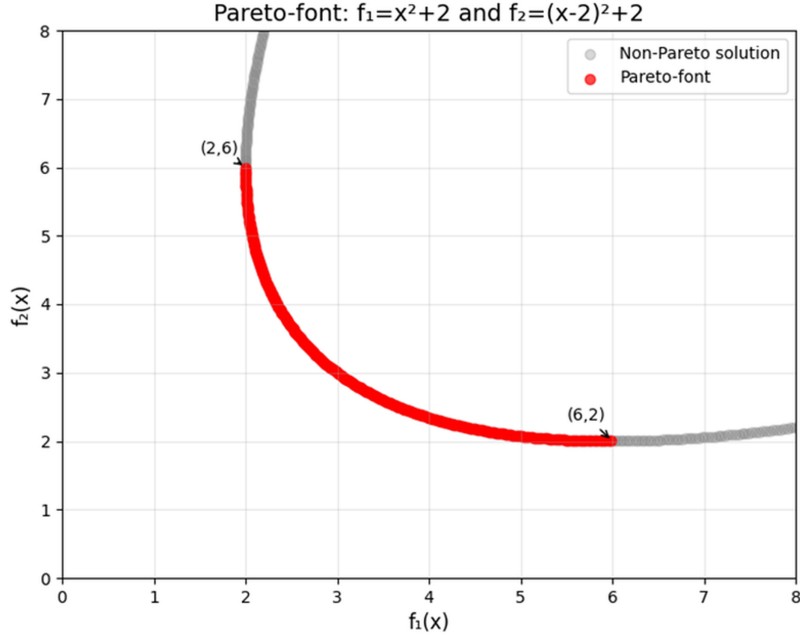

**Figure 3 Pareto front for a bi-objective quadratic optimization problem.**

## Evolution of multi-objective optimization algorithms

The field of MOO has evolved significantly, from early mathematical programming to intelligent, learning-based algorithms that support real-time decision-making. Table 3 outlines five key stages in this progression. Initial methods focused on scalarization and linear transformations, followed by evolutionary and hybrid strategies designed to enhance diversity and convergence. Recent advances integrate machine learning and neuro-evolution, enabling adaptive and context-aware optimization suited to complex urban systems. Each stage reflects a shift in how optimization problems are formulated, solved, and interpreted, driving more autonomous and efficient decision support in smart city applications.

### *Classification of multi-objective optimization algorithms*

MOO algorithms have evolved through interdisciplinary inspiration, drawing from biological evolution, mathematical theory, physical sciences, and artificial intelligence. Bio-inspired algorithms, such as genetic algorithms (GA), particle swarm optimization (PSO), and ant colony optimization (ACO), mimic natural selection, swarm intelligence, and pheromone-based decision-making for global search. Mathematical theory-driven approaches, including the weighted sum method and Multi-Objective Evolutionary Algorithm Based on Decomposition (MOEA/D), leverage mathematical optimization techniques to structure and decompose multi-objective problems. Physics-inspired methods, such as simulated annealing (SA) and gravitational search algorithm (GSA), model optimization as energy minimization or force-driven convergence. With the advancement of artificial intelligence, machine learning-enhanced approaches, like neural

**Table 3 Historical evolution of multi-objective optimization algorithms.**

| Stage | Time period | Core methodologies | Representative algorithms | Contributions & limitations |
|---|---|---|---|---|
| Mathematical programming | 1950s–1980s | Scalarization methods, including weighted sum and $\varepsilon$-constraint; transforming multi-objective problems into sequential single-objective problems | Linear programming, Goal programming | Theoretically strong but relies on prior weight assignment and struggles with non-convex, high-dimensional Pareto fronts (*Charnes & Cooper, 1957*). |
| Emergence of evolutionary algorithms | 1990s–2000s | Introduction of Pareto dominance, non-dominated sorting, and genetic operations for global search | NSGA, SPEA, PAES | Enabled automatic generation of non-dominated solutions but suffered from selection pressure issues in high-dimensional problems (*Srinivas & Deb, 1994*). |
| Hybrid strategy integration | 2000s–2010s | Combination of decomposition strategies, indicator-based selection, and reference-point guidance to enhance convergence and diversity control | NSGA-II, MOEA/D, IBEA | Improved solution set distribution using crowding distance, objective-space decomposition, and hypervolume-based selection, but sensitive to parameter tuning (*Deb et al., 2002a*). |
| Handling complex Pareto fronts | 2010s–2020s | Incorporation of local search, surrogate modeling, and adaptive mechanisms to tackle non-uniform, multimodal, or dynamic Pareto fronts | KnEA, MOPSO, dynamic MOEA | Improved adaptability to complex Pareto fronts but incurred high computational costs (*Zhang, Tian & Jin, 2015*). |
| Automated & intelligent optimization | 2020s–Present | Integration of machine learning techniques (*e.g.*, neuro-evolution, reinforcement learning) for parameter adaptation and real-time optimization | ML-MOEA, Neuro-assisted evolutionary algorithms | Reduced human intervention and enhanced adaptability to dynamic environments, but interpretability remains a challenge (*Galván & Mooney, 2021*). |

**Table 4 Classification of MOO algorithms.**

| Category | Representative algorithms | Fundamental principle | Year |
|---|---|---|---|
| Bio-inspired algorithms | Genetic algorithm (GA) | Simulates natural selection and genetic recombination based on Darwinian evolution | 1975 |
| | Particle swarm optimization (PSO) | Inspired by bird flocking behavior, where particles update their velocity and position based on individual and group best solutions | 1995 |
| | Ant colony optimization (ACO) | Models pheromone-based communication in ant colonies for cooperative pathfinding | 1992 |
| Mathematical theory-driven algorithms | Weighted sum method | Uses linear programming techniques to transform MOO into a single-objective problem | 1950s |
| | MOEA/D | Applies game theory and decomposition strategies to divide high-dimensional objectives into subproblems for cooperative solving | 2007 |
| Physics-inspired algorithms | Simulated annealing (SA) | Maps the annealing process in metallurgy, using a temperature-controlled probabilistic acceptance mechanism to avoid local optima | 1983 |
| | Gravitational search algorithm (GSA) | Based on Newton's law of gravitation, simulating the attraction forces between solutions to guide convergence | 2009 |
| Machine learning-enhanced optimization | Neural network-based surrogate models | Uses deep learning to approximate objective functions, reducing computational costs for real-world optimization | 2010s |
| | Reinforcement learning-based optimization | Integrates Markov decision processes (MDP) and vectorized reward functions for optimizing strategies in dynamic environments | 1998 |

network surrogate models and reinforcement learning-based optimization, integrate deep learning and dynamic decision-making to improve computational efficiency and adaptability. These algorithmic categories, along with their foundational principles and historical development, are summarized in Table 4.

# FINDINGS OF THE SLR

## SLR overview

Figure 4 presents the key statistical characteristics derived from this SLR, structured across four analytical dimensions to provide a comprehensive understanding of MOO applications in smart cities. As shown in Fig. 4A, shows a steady increase in publications from 2015, peaking in 2024. The upward trend is further emphasized in Fig. 4B, where a sharp acceleration in recent years culminates in a peak of 36 articles in 2024. Figure 4C illustrates the geographic distribution of corresponding authors, with a concentration of studies from a few leading countries. Finally, Fig. 4D shows the proportional distribution of publications by year, reinforcing the increasing academic engagement with MOO-related research in smart city contexts. Collectively, these visualizations reflect the field's growing maturity, geographic focus, and rapid expansion.

## Primary smart city scenarios

Figure 5 presents six primary smart city scenarios addressed by MOO: infrastructure (31.6%), energy (20.5%), transportation and mobility (18.8%), IoT and cloud computing (16.2%), agriculture and ecological management (7.7%), and water resource management (5.1%). Infrastructure, energy, and transportation collectively dominate, accounting for over 70% of reviewed studies, highlighting their central role in urban optimization. Agriculture and water management, despite their critical role in sustainability, remain underrepresented.

Figure 6 further shows strong interconnections among infrastructure, energy, transportation, and IoT systems, indicating frequent co-optimization. In contrast, agriculture and water resource domains show limited cross-domain integration, aligning with their lower representation in existing research.

Overall, the findings suggest a clear opportunity to enhance integration and sustainability optimization, particularly within agriculture and water management scenarios.

### *Mapping objectives, domains, and constraints for smart city MOO*

To enhance scope clarity and align MOO with urban priorities, Fig. 7 presents a structured conceptual framework. This visualization organizes the review around three key components: Fig. 7A core optimization objectives and their expected benefits, Fig. 7B major urban sectors where MOO techniques are applied, and Fig. 7C societal and governance constraints that influence model design and outcomes. Rather than listing algorithms by domain, the framework emphasizes both the technical applications of MOO

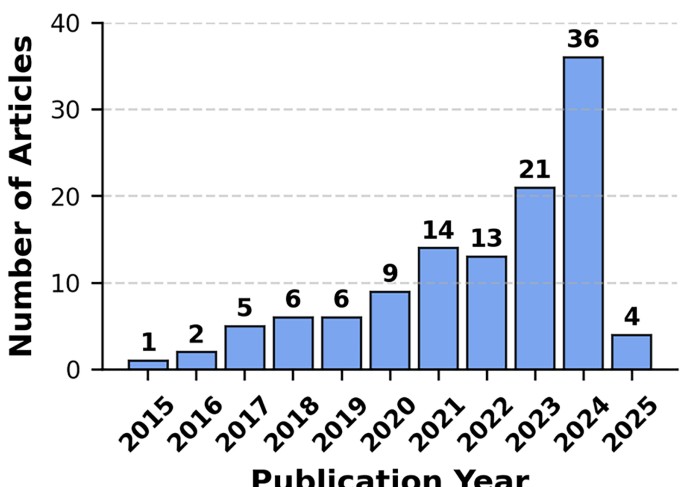

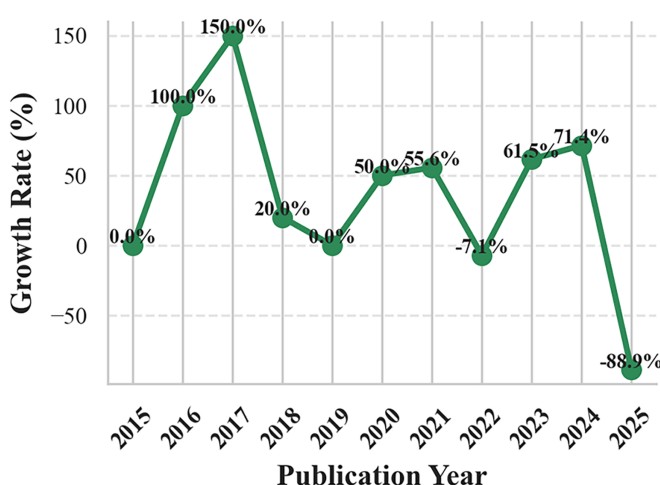

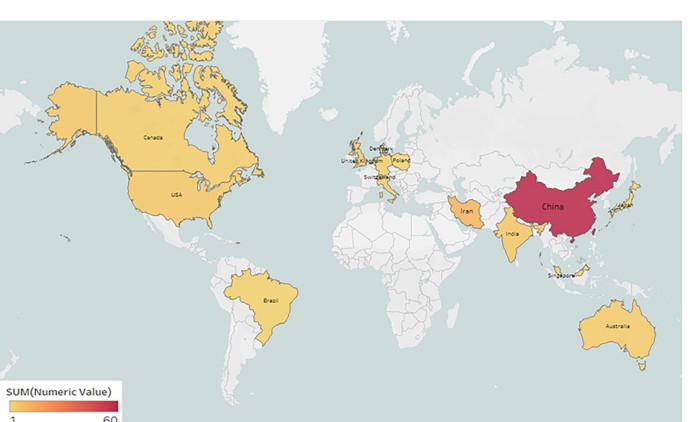

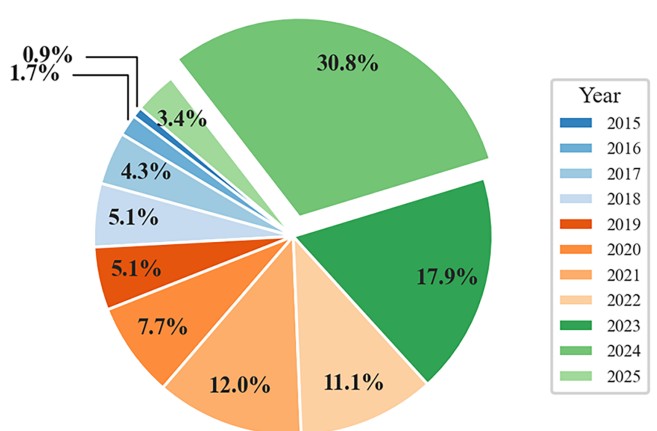

**Figure 4 SLR overview.** (A) Annual number of publications (2015–2025). (B) Year-on-year growth rate of publications. (C) Country distribution of corresponding authors; darker shade = more papers. (D) Percentage share of publications by year.

and their strategic relevance in smart city contexts. It provides a high-level entry point that integrates environmental goals, operational challenges, and social considerations, offering a holistic perspective on how optimization interacts with real-world urban systems.

## Comparative analysis of multi-objective optimization algorithms in smart city scenarios

This section provides a comparative analysis of MOO algorithms across diverse smart city scenarios. It covers algorithm classifications and scenario suitability, strategies for resolving objective conflicts, performance evaluations, and computational considerations, ensuring a cohesive examination from theoretical categories to practical outcomes.

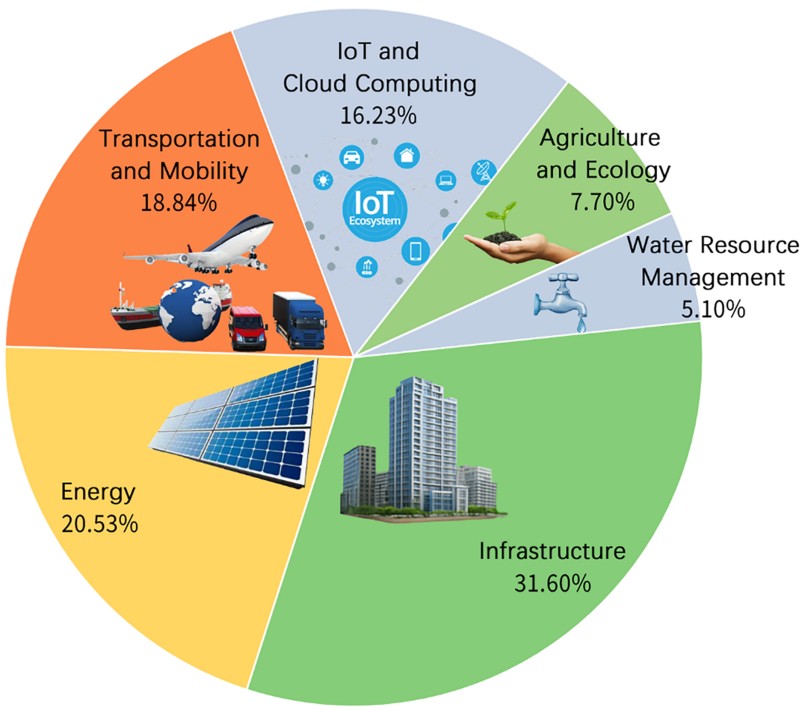

**Figure 5 Scenario distribution.**

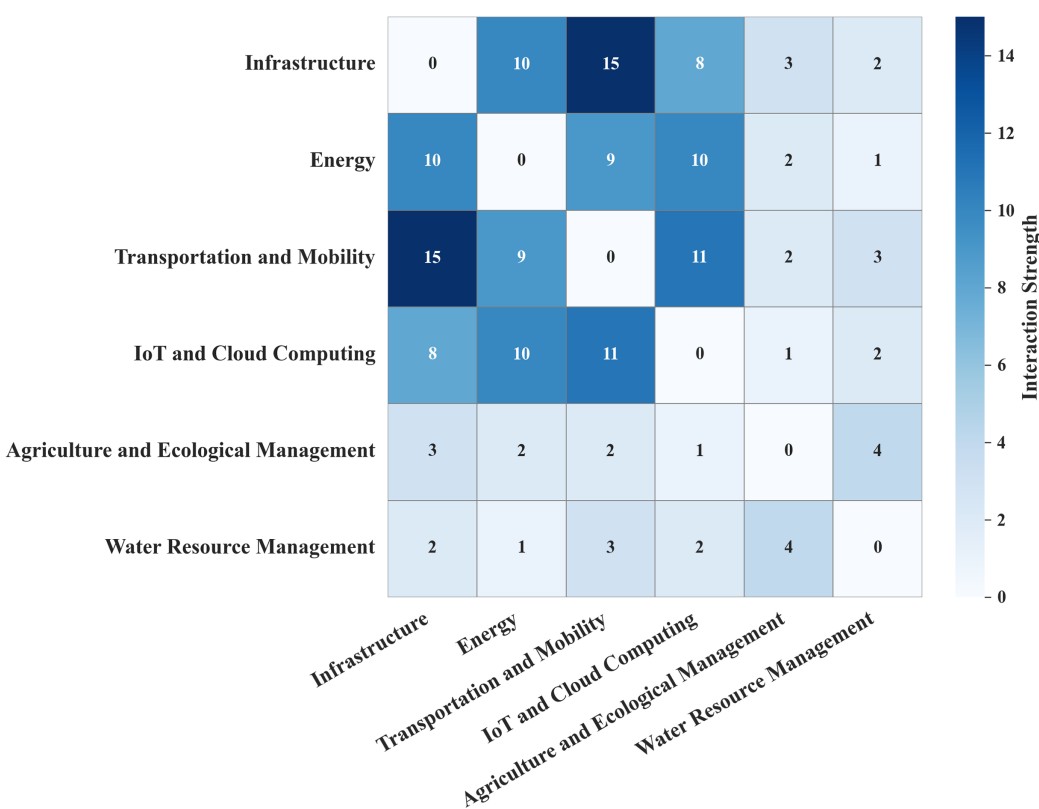

**Figure 6 Scenario interaction matrix.**

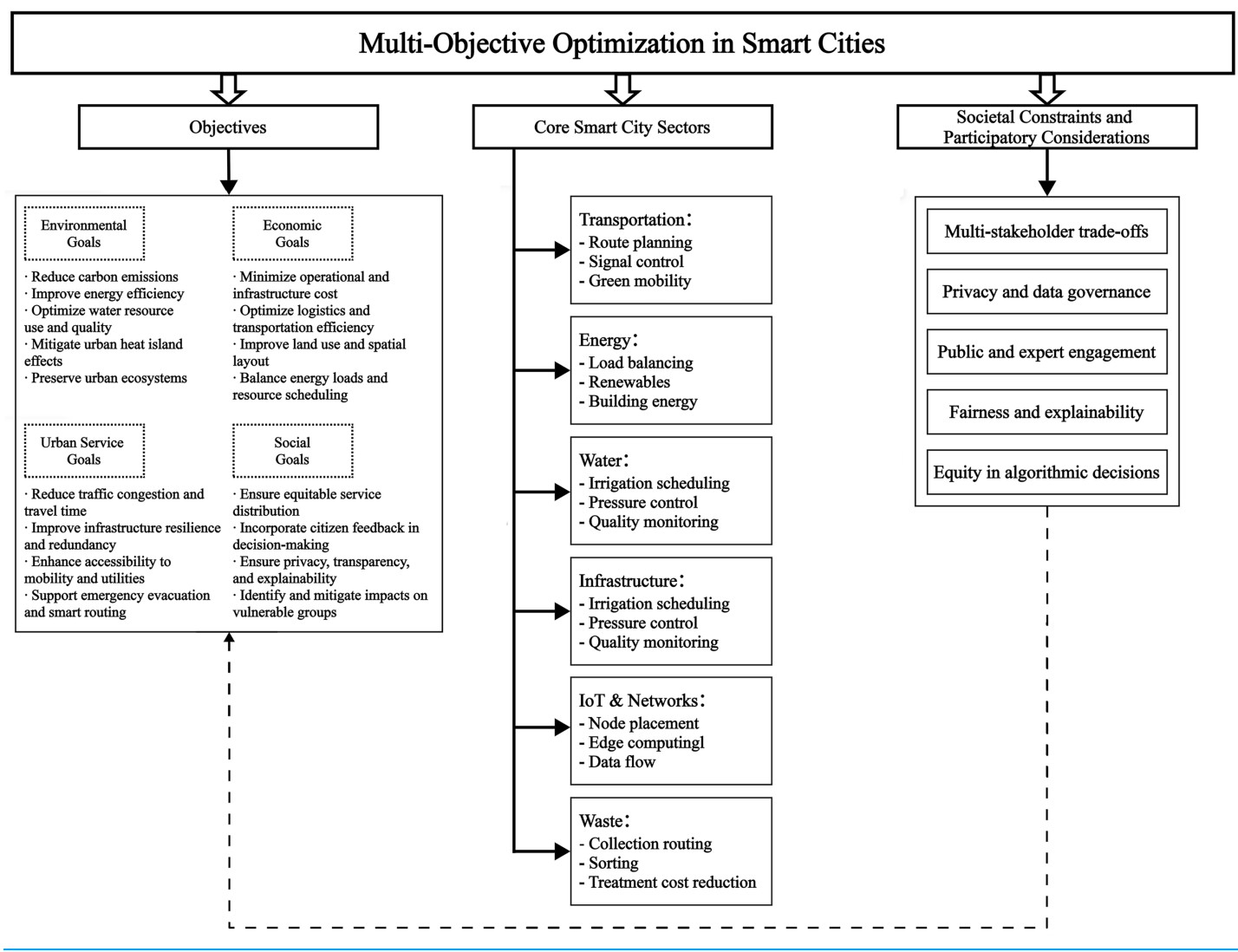

Figure 7 Layered framework of MOO in smart cities.   

### Algorithmic classification and scenario suitability

The classification of MOO algorithms for smart city applications depends on both their theoretical foundations and empirical performance across various urban scenarios. A clear classification framework helps reveal how each algorithm addresses complex urban challenges involving multi-dimensional trade-offs. For instance, NSGA-II has been applied to edge server placement in smart cities, improving infrastructure deployment and energy efficiency (*Zhao et al., 2021*). Similarly, a multi-objective genetic algorithm optimized vehicle routing in green city initiatives, enhancing urban transportation systems (*Zhao, Bian & Mei, 2024*).

**Nature-inspired algorithms (NIA).** This category represents the largest class of MOO algorithms, including evolutionary algorithms (EA) (*e.g.*, NSGA-II, NSGA-III), swarm intelligence (SI) methods (*e.g.*, PSO, ACO), and other techniques like gray wolf optimizer

(GWO), firefly algorithm (FA), bat algorithm (BA). These algorithms mimic natural or physical processes through stochastic search, enabling flexible exploration of complex, nonlinear solution spaces. EA techniques demonstrate robust convergence in infrastructure planning and energy optimization problems (*Maleki et al., 2022*; *Reitberger et al., 2024*). However, their high computational demands necessitate distributed computing for scalability. SI algorithms excel in dynamic, real-time scenarios such as traffic flow optimization and IoT network management due to their adaptability (*Sajid et al., 2021*; *Panda, Muthuraman & Elsts, 2024*). Yet, they risk premature convergence and may require adaptive parameter tuning or hybridization. Other NIA approaches like GWO and FA have proven effective in resource allocation and traffic management but depend on robust feedback mechanisms to handle dynamic conditions (*Othman, Darwish & Abd El-Moghith, 2023*; *Kazmi et al., 2025*).

**Physics-based and model-driven algorithms.** These methods use deterministic modeling based on physical laws, contrasting with the stochastic nature of NIA. For example, energy flexibility was improved in a building–vehicle sharing network using optimization modeling (*Zhou et al., 2020*), and urban microclimates were co-designed with buildings for thermal optimization (*Zhao, Li & Wang, 2025*). These methods are suitable for energy transfer analysis, infrastructure design, and thermal regulation tasks (*Wu et al., 2024*; *Shafiq et al., 2024*). Their main strength lies in consistent, high-fidelity results crucial for smart city planning, though real-time scalability may be hindered by computational intensity.

**Machine learning-assisted optimization.** Integrating machine learning with MOO adds adaptability and predictive capabilities. For instance, spatial projection pursuit using geospatial data was employed to assess urban vitality (*Zhang et al., 2024b*). Deep learning enables modeling of complex, nonlinear patterns, supporting adaptive planning in logistics, energy systems, and land use (*Preuveneers, Tsingenopoulos & Joosen, 2020*; *Zhang et al., 2024a*). While effective in pattern recognition and predictive modeling, these approaches require computational resources and benefit from lightweight neural architectures and transfer learning.

**Hybrid frameworks.** These combine NIA's exploratory abilities with machine learning's predictive accuracy. For example, hybrid frameworks have been applied in fog–cloud computing environments requiring real-time responsiveness (*Masoumi & van Genderen, 2024*; *Mokni & Yassa, 2024*). By combining evolutionary search and data-driven models, they deliver both adaptability and efficiency. Yet, integration complexity remains a challenge, calling for modular and scalable system designs.

**Cross-domain optimization strategies.** These strategies synchronize multiple urban systems such as transportation, energy, and communication to work in concert. Coordinating across domains is pivotal for improving overall system efficiency and resilience (*Stoyanova & Monti, 2019*). For instance, optimizing network topology

improved infrastructure layout (*Zhou et al., 2019*), enabling better service coordination. Such strategies are crucial in addressing the intertwined nature of urban problems and ensuring system synergy.

In summary, nature-inspired algorithms offer adaptability for dynamic problems, while physics-based and model-driven methods provide reliable results for critical infrastructure. Machine learning enhances adaptability and scalability in predictive tasks. Hybrid models combine strengths from both domains, and cross-domain strategies ensure holistic system integration. For instance, *Zhang et al. (2024a)* developed an adaptive MOO traffic signal model balancing efficiency and energy, exemplifying integrated algorithmic design for real-time urban systems.

### Objective conflicts and resolution approaches

Resolving conflicting objectives is central to MOO in smart cities, where trade-offs between sustainability, efficiency, cost, and adaptability frequently arise. Based on our review of 117 studies, Fig. 8 visualizes the distribution of objective counts, showing that most MOO tasks fall into three categories of dimensional complexity: low, mid, and high. Each category presents unique modeling and computational challenges (*Ishibuchi, Tsukamoto & Nojima, 2008*). Detailed evaluation criteria and benchmark functions relevant to these conflict levels are discussed in subsequent sections to avoid redundancy.

### Low-dimensional conflicts: rapid convergence and high adaptability

Low-dimensional conflicts (2–3 objectives) are frequently observed in energy systems, transportation networks, and smart building optimization. In scenarios balancing energy efficiency, thermal comfort, and operational cost, NSGA-II demonstrated robust convergence, achieving an 18% reduction in energy use while preserving indoor comfort and economic feasibility (*Lotfi & Hassan, 2024*).

In traffic optimization, PSO with adaptive inertia weight strategies simultaneously minimized travel time, emissions, and fuel consumption. Results reported include a 20% reduction in energy use and an 8% decrease in average travel time (*Liu et al., 2023a*). Despite its effectiveness, PSO's performance deteriorates in non-convex landscapes, often requiring hybridization.

### Mid-dimensional conflicts: balancing complexity, scalability, and convergence

With 4–10 objectives, mid-dimensional problems often arise in integrated land-use, transportation, and ecosystem planning. A hybrid NSGA-III + MOEA/D framework reduced ecosystem service loss by 15%, enhanced transportation accessibility by 10%, and maintained economic viability (*Maleki et al., 2022*). These algorithms effectively handle the increasing diversity and decision complexity inherent in urban interdependencies.

Further, combining NSGA-III with GWO and deep neural networks (DNNs) has proven effective in fog/IoT settings, improving resource utilization by 30% and reducing latency by 25% (*Mokni & Yassa, 2024*).

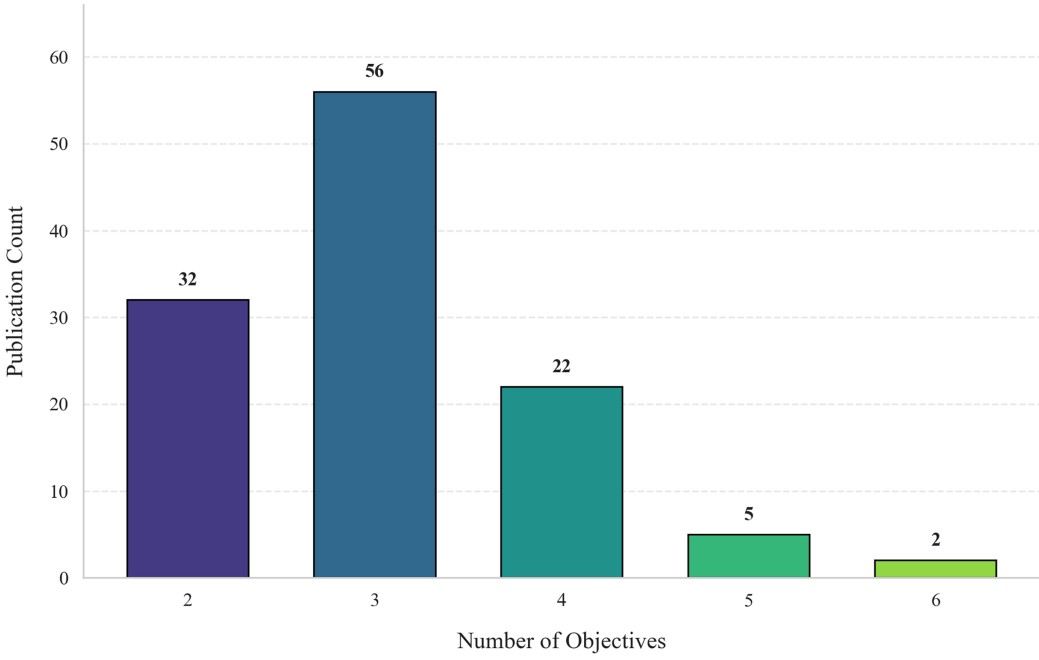

**Figure 8 Distribution of publications by multi-objective complexity.**

### High-dimensional conflicts: underexplored but computationally demanding

High-dimensional conflicts (more than 10 objectives) remain underexplored due to the complexity of dominance preservation and solution diversity. New methods such as many-objective grasshopper optimization algorithm (MaOGOA) (*Kalita et al., 2024*) and MaOEA/TS (*Zhao et al., 2024*) address these limitations by integrating adaptive reference points and staged optimization. These approaches demonstrate promising results on test suites like Many-objective Test Function suite (MaF) and large-scale multi objective optimization problems (LSMOP) but are rarely applied in real-world smart city scenarios.

To operationalize high-dimensional optimization in practice, future work must balance scalability, interpretability, and convergence robustness, especially when trade-offs span multiple subsystems.

### Evaluation criteria and benchmark resources in smart city MOO

To ensure fair comparison and reproducibility in MOO research, particularly in the diverse and dynamic context of smart cities—standardized evaluation criteria and benchmark resources are essential. This section integrates geometric considerations of Pareto fronts, performance metrics, benchmark functions, and real-world datasets to form a cohesive foundation for rigorous algorithm assessment.

**1. Geometric characteristics and optimization implications.** The structure of the Pareto front, whether known (*e.g.*, convex or linear) or unknown (*e.g.*, multimodal or disconnected), influences the effectiveness of optimization strategies. When the front is analytically known, mathematical techniques like the Karush–Kuhn–Tucker (KKT)

**Table 5 Key quality evaluation metrics for Pareto-optimal solution sets.**

| Metric | Definition and computation | Suitability in urban MOO tasks |
| --- | --- | --- |
| Hypervolume (HV) | Measures the dominated space between the solution set and a reference point. | Captures convergence and diversity, though computationally expensive in high dimensions (*Zitzler & Thiele, 1998*). |
| Inverted generational distance (IGD) | Distance between true Pareto front and obtained set. | Reflects convergence accuracy (*Bezerra, López-Ibáñez & Stützle, 2017*). |
| Spacing | Measures the evenness between neighboring solutions. | Indicates uniform distribution but not convergence (*Scheepers & Engelbrecht, 2016*). |
| Spread | Ratio of extreme distance to average spread. | Reflects coverage of entire front (*Deb et al., 2002b*). |

conditions or ε-constraint methods are employed. However, in complex urban systems, the front is often unknown, requiring evolutionary strategies that use non-dominated sorting, diversity maintenance, and adaptive parameter control. Table 5 summarizes the core evaluation metrics commonly used to assess the quality of Pareto-optimal solution sets in such scenarios.

**2. Benchmark function suites for algorithm testing.** Benchmark functions provide standardized environments to validate algorithm scalability, robustness, and adaptability. While ZDT, DTLZ, and WFG are often associated with low-, mid-, and high-dimensional test cases respectively, these suites are in fact highly configurable. Depending on the design of experimental studies, each can be adapted to various objective counts and Pareto front complexities (*Deb et al., 2002b*; *Huband et al., 2006*). Table 6 summarizes the typical characteristics of these dimensional categories along with representative benchmark functions commonly used in the literature.

**3. Real-world datasets in smart city domains.** To support the development and validation of MOO algorithms under realistic conditions, a range of domain-specific datasets have been employed across smart city research. These datasets enable algorithm testing under real-world constraints, capturing the complexity, noise, and multi-stakeholder objectives inherent to urban systems.

- **Transportation:** The *METR-LA* and *PeMS* datasets, collected from California highway sensors, are widely used in traffic forecasting and dynamic routing optimization (*He, 2025*; *Caltrans Division of Traffic Operations, 2024*). Additionally, the *NYC Taxi Trip* dataset supports large-scale mobility modeling and demand-responsive scheduling (*New York City Taxi and Limousine Commission, 2024*).
- **Energy systems:** The *Pecan Street* dataset provides high-resolution electricity consumption data at the household level, supporting demand response and smart grid optimization (*Pecan Street Inc., 2024*). The *UMass Smart** dataset offers microgrid and appliance-level energy traces for distributed energy management research (*University of Massachusetts Amherst, 2024*).
- **Water management:** Hydrological modeling tools such as *SWAT* and *EPA SWMM* offer synthetic and empirical datasets for simulating stormwater drainage, flood risk, and infrastructure resilience (*U.S. Department of Agriculture, 2024*; *Rossman & Simon, 2022*).

**Table 6 Analysis of dimensional complexity in MOO.** Function assignments are based on common usage patterns, not strict dimensional constraints.

| Dimension | Problem characteristics | Representative benchmark functions |
|---|---|---|
| Low (2–3) | Simple geometric structure of the Pareto front (convex, concave, or disconnected); easy to visualize and analyze | ZDT Series (*Zitzler, Deb & Thiele, 2000*), Kursawe Function (*Deb et al., 2002a*) |
| Medium (4–10) | Emergence of the curse of dimensionality; sparse distribution of solutions; reduced search efficiency | DTLZ Series (*Deb et al., 2002b*), WFG Toolkit (*Huband et al., 2006*) |
| High (>10) | Severe objective redundancy; loss of dominance pressure; need for dimensionality reduction or adaptive reference-based strategies | MaF Series (*Zhang, Liu & Yao, 2023*), MAOP, LSMOP (*Kalita et al., 2024*) |

The *WaterTAP* dataset, curated by the U.S. Department of Energy, facilitates the optimization of water treatment and distribution systems (*Atia et al., 2024*).

- **Cross-domain applications:** Several benchmark initiatives, such as *CityLearn* and *IEEG Smart City Challenge*, integrate multiple urban subsystems (*e.g.*, energy–mobility–comfort) to evaluate algorithm adaptability in coupled environments, bridging the gap between siloed research and real-world complexity.

These datasets serve as crucial resources for validating algorithm robustness and transferability, enabling MOO studies to move beyond theoretical constructs and address pressing urban challenges with real-time and multi-objective constraints. Together, these standardized resources form a methodological bridge between algorithm development and smart city deployment, fostering reproducible, scalable, and domain-adaptive MOO research.

### Algorithmic strengths, weaknesses, and selection criteria

**Bio-inspired algorithms.** Bio-inspired algorithms have emerged as a dominant paradigm in MOO for smart city applications, driven by their population-based exploration capabilities and robust performance in complex, nonlinear, and dynamic environments. These algorithms have demonstrated high adaptability in diverse urban tasks such as land-use allocation, energy system control, transportation scheduling, and computational resource distribution. However, their performance can deteriorate in high-dimensional settings due to convergence slowdowns, increased computational overhead, and limited ability to preserve the diversity of solutions.

To interpret Table 7, we classify the performance of bio-inspired MOO algorithms across three core dimensions: convergence, diversity, and computational complexity, based on both structural characteristics and benchmark results reported in the literature.

In terms of convergence, NSGA-III (*Seada, Abouhawwash & Deb, 2016*; *Gupta & Nanda, 2019*) exhibits superior performance, primarily due to its reference direction—based niche preservation mechanism. *Gupta & Nanda (2019)* reported inverted generational distance (IGD) values as low as 0.005 across 3- to 8-objective benchmark problems, underscoring its high convergence accuracy. Similarly, Emotion Driven Monocular Face Capture and Animation (EMOCA) (*Othman, Darwish & Abd El-Moghith, 2023*) achieved strong convergence results in IoT routing scenarios, with fault tolerance reaching 0.98 under large-scale deployment conditions. In contrast, NSGA-II

**Table 7 Quantitative comparison of core performance metrics in bio-inspired MOO algorithms.**

| Algorithm | Convergence | Diversity | Computational complexity | Supporting references |
|---|---|---|---|---|
| NSGA-II | High | Medium | Medium | *Sato, Sato & Miyakawa (2017)*, *Li et al. (2022a)* |
| MOEA/D | Medium | High | Low | *Alagumathi & Thangavelu (2024)*, *Fu et al. (2021)* |
| SPEA2 | Medium | High | High | *Liu & Zhang (2019)*, *Jiang & Yang (2015)* |
| NSGA-III | High | High | High | *Seada, Abouhawwash & Deb (2016)*, *Gupta & Nanda (2019)* |
| PSO | Medium | Medium | Low | *Liu, Li & Zhu (2021)*, *Harron, Saxena & Kumari (2024)* |
| EMOCA | High | High | High | *Othman, Darwish & Abd El-Moghith (2023)* |
| ACO | Medium | Medium | High | *Chitty (2018)*, *Sabery, Danishyar & Mubarez (2023)* |

(*Sato, Sato & Miyakawa, 2017*; *Li et al., 2022a*) and SPEA2 (*Liu & Zhang, 2019*) demonstrate moderate convergence. For instance, *Li et al. (2022a)* employed SVR-TOPSIS analysis and found that NSGA-II achieved acceptable but suboptimal Pareto front accuracy. The standard Strength Pareto Evolutionary Algorithm 2 (SPEA2) implementation delivered consistent results but was outperformed by its improved variants on ZDT3 and DTLZ2 test suites (*Liu & Zhang, 2019*). ACO (*Chitty, 2018*; *Sabery, Danishyar & Mubarez, 2023*), while competitive in simpler contexts, reported up to 9.4% deviation from optimal solutions on TSPLIB instances, reflecting limited convergence capability in high-dimensional settings.

With respect to *diversity*, MOEA/D (*Fu et al., 2021*; *Alagumathi & Thangavelu, 2024*), NSGA-III (*Seada, Abouhawwash & Deb, 2016*), and EMOCA (*Othman, Darwish & Abd El-Moghith, 2023*) demonstrate a consistently strong ability to maintain well-distributed solution sets. *Fu et al. (2021)* introduced adaptive weight vector strategies in MOEA/D-AW, yielding significantly improved distribution on WFG benchmarks. *Seada, Abouhawwash & Deb (2016)* further showed that local search integration into NSGA-III enhances diversity preservation in complex Pareto landscapes. EMOCA integrates crowding distance evaluation throughout all evolutionary phases, effectively maintaining near-uniform spacing among non-dominated solutions. In contrast, NSGA-II, PSO, and ACO typically offer only moderate diversity maintenance. For example, *Liu, Li & Zhu (2021)* observed that standard PSO is susceptible to swarm stagnation, although diversity can be improved using biologically inspired variants such as SSI-PSO. Similarly, *Chitty (2018)* noted that ACO suffers from local optima entrapment without explicit diversity enhancement mechanisms.

Regarding computational complexity, PSO and MOEA/D are widely recognized for their computational efficiency. PSO operates with linear time complexity relative to particle count and dimensionality (*Liu, Li & Zhu, 2021*), making it suitable for real-time applications and large-scale problem settings. MOEA/D (*Alagumathi & Thangavelu, 2024*), leveraging subproblem decomposition and neighborhood-based update schemes, avoids the computational burden associated with nondominated sorting ($O(MN^2)$) that is characteristic of NSGA-II and NSGA-III. On the other hand, NSGA-III (*Gupta & Nanda, 2019*), SPEA2 (*Jiang & Yang, 2015*), and EMOCA (*Othman, Darwish & Abd El-Moghith, 2023*) are associated with high computational demands. *Jiang & Yang (2015)* estimated

**Table 8 Composite star ratings of algorithms across smart city domains.**

| Algorithm | Infrastructure | Energy | Water resources | Transportation | IoT/Network |
|---|---|---|---|---|---|
| ACO | ★ | ★ | ★ | ★ | ★★ |
| EMOCA | ★ | ★ | ★ | ★ | ★★★ |
| MOEA/D | ★ | ★ | ★ | ★★ | ★ |
| NSGA-II | ★★★★★ | ★★★★★ | ★★★★ | ★★★★★ | ★★★ |
| NSGA-III | ★★ | ★★ | ★ | ★★ | ★ |
| PSO | ★★★ | ★★★ | ★★★ | ★★★ | ★★★★ |
| SPEA2 | ★★ | ★★★ | ★★ | ★★★ | ★★ |

**Note:**
Stars represent composite scores based on two dimensions: (i) literature frequency and (ii) performance suitability. Ratings are interpreted as follows: ★★★★★ (frequent use and strong suitability), ★★★★ (broad applicability with solid performance), ★★★ (moderate use and acceptable alignment), ★★ (limited use or partial mismatch), and ★ (rare use or poor fit). These star ratings are intended as structured summaries of suitability trends across domains, rather than definitive rankings of algorithmic quality. *Score-to-star mapping:* 6 points = ★★★★★, 5 = ★★★★, 4 = ★★★, 2–3 = ★★, 0–1 = ★.

that SPEA2's k-nearest neighbor–based density estimation results in a worst-case complexity of $O(MN^3)$. NSGA-III's reference-point association and EMOCA's multi-phase crowding strategies further increase runtime overhead, which may limit scalability in time-sensitive urban applications.

This comparative synthesis enables a more nuanced understanding of the trade-offs among convergence reliability, diversity maintenance, and computational feasibility. It offers practitioners a decision-support reference for selecting appropriate bio-inspired MOO algorithms tailored to domain-specific constraints and resource availability.

To systematically evaluate the domain-specific applicability of bio-inspired MOO algorithms, a structured scoring framework was developed to integrate both empirical usage patterns and algorithmic performance characteristics. The results are summarized in Table 8, which presents a composite star-rating matrix across major smart city domains. Each algorithm—domain pair was assigned a composite score ranging from 0 to 6, based on two components. This 0–6 range ensures sufficient granularity while maintaining interpretability when translated into a five-level star system.

- **(i) Literature frequency (0–3 points):** Quantified by the number of reviewed studies (out of 117) that applied the algorithm in the corresponding smart city domain:

  - 10 or more studies: 3 points
  - 5 to 9 studies: 2 points
  - 1 to 4 studies: 1 point
  - 0 studies: 0 points

- **(ii) Performance suitability (1–3 points):** Assigned based on the algorithm's convergence capability, solution diversity, and computational complexity, as summarized in Table 7. Specifically:

- 3 points: "High" in two or more dimensions, or "High" in one and "Medium" in the others
- 2 points: combination of "Medium" and one "High" or one "Low"
- 1 point: two or more "Low" scores, or dominant "High" complexity without offsetting strengths

The combined score (0–6) was then translated into a five-level star rating to visualize algorithm suitability across domains.

These comparative insights are further supported by empirical applications across smart city domains. For instance, NSGA-II remains a dominant choice in land-use and environmental planning due to its reliable convergence and manageable complexity in low-dimensional tasks (*Deb et al., 2002a*; *Liu et al., 2023b*). Its successful deployment extends to blockchain-based smart contract testing (*Alkhazi & Alipour, 2023*), flood mitigation planning (*Lin, Sun & Nijhuis, 2024*), and stormwater infrastructure design, where it improved runoff control by 20% (*Liu et al., 2023b*). However, as the number of objectives increases, NSGA-II suffers from declining diversity and slower convergence, as noted by *Feng et al. (2021)*. In contrast, NSGA-III and EMOCA demonstrate superior diversity maintenance and scalability in high-dimensional contexts (*Maleki et al., 2022*; *Othman, Darwish & Abd El-Moghith, 2023*), aligning with their high performance scores in Table 7.

MOEA/D, by decomposing the problem into scalar subproblems, achieves low computational overhead and has proven efficient for embedded applications such as IoT node placement and real-time transportation scheduling (*Zhang & Li, 2007*; *Wang, Jin & Doherty, 2015*). MOGA has shown superior performance in resource allocation, achieving optimal trade-offs in logistics and land use (*Li et al., 2024*), while NSGA-II remains effective in route optimization under emission constraints (*Sajid et al., 2021*).

In network-centric environments, EMOCA excels due to its crowding-based diversity control, enabling a 23% reduction in communication latency and enhanced fault resilience in IoT sensor networks (*Othman, Darwish & Abd El-Moghith, 2023*). PSO and ACO are favored for their low-latency convergence in dynamic routing and scheduling tasks, such as waste collection and service placement (*Kazmi et al., 2025*; *Ahmad et al., 2020*), though both are prone to premature convergence in complex landscapes.

Energy-focused applications highlight further trade-offs: NSGA-II has reduced $CO_2$ emissions by 7.5% in building-vehicle energy-sharing networks (*Zhou et al., 2020*) and improved fire safety in ventilation optimization (*Xu et al., 2024*), yet remains constrained by high computational demands, as seen in hydroelectric optimization (*Uen et al., 2018*).

To address such limitations, adaptive variants have emerged. Adaptive MOGA introduces self-adjusting mutation control to improve convergence in vehicle routing (*Zhao et al., 2024*); Hybrid PSO enhances farmland conservation with improved search capabilities but increased runtime (*Wang et al., 2020*); and Dynamic PSO incorporates inertia weight adjustment for improved stability in real-time optimization (*Wang, 2020*).

To synthesize the comparative evidence across algorithm families, Table 9 consolidates key algorithm classes with their typical domains, primary strengths and weaknesses, and

**Table 9 Bio-inspired optimization algorithms for smart city applications.**

| Algorithm | Domain | Strengths | Limitations | Main performance and representative cases |
|---|---|---|---|---|
| NSGA-II, NSGA-III | Land-use, energy, traffic control | Strong diversity; efficient Pareto front approximation | High computation; poor adaptability to dynamic settings | 31.3% farmland loss reduction; enhanced diversity (*Shirzadi Babakan & Taleai, 2015*; *Schwaab et al., 2018*; *Uen et al., 2018*; *Zhou et al., 2020*; *Feng et al., 2021*; *Saha et al., 2021*; *Maleki et al., 2022*; *Wang, Han & De Vries, 2022*; *Liu et al., 2023b*; *Lin, Sun & Nijhuis, 2024*; *Yang et al., 2024*; *Xu et al., 2024*) |
| MOGA | Logistics, resource allocation | Balances multi-dimensional constraints | Parameter tuning complexity | 17% logistics cost reduction (*Sajid et al., 2021*; *Li et al., 2024*) |
| EMOCA | IoT, sensor networks | Reduced latency; improved fault tolerance | Computationally intensive; limited deployment | 23% latency reduction; better fault resilience (*Othman, Darwish & Abd El-Moghith, 2023*) |
| PSO, ACO | Scheduling, dynamic routing | Real-time efficiency; low latency | Prone to early convergence; scalability issues | 20.4% latency cut (*vs.* GA); 12.5% waste collection cost saving (*Ahmad et al., 2020*; *Maleki et al., 2022*; *Liu et al., 2023b*; *Lin, Sun & Nijhuis, 2024*; *Kazmi et al., 2025*) |
| Hybrid PSO, Adaptive MOGA | Traffic, land conservation | Adaptive mutation improves search capability | Long runtime; instability under complex constraints | 16.8% travel time reduction (*Wang, 2020*; *Zhao, Bian & Mei, 2024*) |

representative quantitative outcomes. This table complements the prior two by offering more fine-grained, case-level guidance for real-world MOO problem settings.

In summary, while no single algorithm demonstrates universal superiority, this integrated evaluation highlights how convergence-diversity tradeoffs, computational constraints, and real-world objectives guide the practical selection of bio-inspired algorithms in smart cities. Future research should aim to enhance hybridization strategies and improve adaptability in dynamic and high-dimensional scenarios.

**Mathematical optimization methods.** Mathematical optimization methods are widely used in task scheduling, energy management, logistics, land use, and water resource management in smart cities to address multi-objective conflicts. Traditional techniques, such as mixed-integer programming (MIP), MOEA/D, and the weighted sum method (WSM), provide rigorous constraint handling and high computational precision. However, their computational complexity and limited adaptability in dynamic environments have led researchers to explore hybrid approaches and machine learning (ML)-enhanced optimization for improved efficiency and flexibility.

In task scheduling and energy management, optimization must balance execution efficiency, energy consumption, and carbon emissions while adapting to changing conditions. *Abdel-Basset et al. (2021)* proposed a Pareto-optimal strategy to reduce computational load and improve throughput. However, as a static optimization method, it struggles with real-time demand fluctuations. *Zhao et al. (2023)* applied WSM to balance cost, load shedding, and renewable energy integration, improving cost efficiency but highlighting challenges in economic feasibility and system resilience. A similar limitation is observed in energy management, where *Algieri, Morrone & Bova (2020)* employed MIP for CHP system optimization, achieving 7.5% higher efficiency and 15% increased energy self-

consumption. Yet, MIP lacks adaptability to fluctuating energy demand, necessitating more flexible strategies.

To address this, machine learning-enhanced optimization has been introduced. *Li et al. (2022b)* proposed a data-driven model for energy management, reducing dependency on precise mathematical modeling while improving adaptability. Similarly, *Pinki Kumar et al. (2025)* combined fuzzy MOO and multi-criteria decision-making to develop an adaptive scheduling strategy. Unlike (*Abdel-Basset et al., 2021*), which relies on fixed optimization weights, this approach dynamically adjusts parameters, improving flexibility at the cost of higher computational demands.

Urban logistics and energy optimization also show an evolution in optimization strategies. *Jiao & Zhang (2025)* applied a gradient-based optimization approach to optimize logistics distribution and land use, significantly reducing transportation costs and carbon emissions. However, as a local optimization method, it is prone to converging to suboptimal solutions in complex multi-objective conflicts. However, as a local optimization method, it is prone to converging to suboptimal solutions in complex multi-objective conflicts.

Table 10 summarizes the representative mathematical optimization algorithms applied in smart city scenarios, highlighting their respective domains, advantages, limitations, and quantitative performance outcomes.

**Physics-based optimization.** Physics-based optimization plays a critical role in MOO, particularly in the domains of urban energy management and heat transfer. Computational fluid dynamics (CFD) is widely adopted to optimize urban airflow and thermal energy distribution (*Cao et al., 2024*). While CFD offers high-fidelity simulation and accuracy, its computational demands severely limit its use in real-time applications. As an alternative, data-driven thermal modeling has been explored to enhance computational efficiency. For instance, *Shafiq et al. (2024)* developed a CNN–LSTM model for HVAC system optimization, achieving 15.7–22.3% gains in energy efficiency and 21.8–28.5% improvements in occupant comfort. However, while such machine learning models improve feasibility, they often sacrifice interpretability and precision, thereby highlighting the trade-off between accuracy and computational practicality.

Beyond heat transfer, cross-domain optimization frameworks that integrate district heating, electricity, and water distribution networks are increasingly applied to improve overall urban energy efficiency. Nevertheless, these frameworks impose significant computational burdens, which has motivated the adoption of metaheuristic physics-inspired algorithms such as simulated annealing (SA) and gravitational search algorithm (GSA).

SA and GSA, both rooted in physical laws, have found extensive application in smart city optimization. SA mimics thermodynamic annealing processes and has demonstrated effectiveness in logistics and routing tasks. For example, *Salehi-Amiri et al. (2022)* employed SA for waste management optimization, yielding notable improvements in route efficiency. On the other hand, GSA, inspired by Newtonian gravitation, has shown promise in communication and network-related problems. *Okafor et al. (2024)* demonstrated that

**Table 10 Mathematical theory-driven optimization algorithms for smart city applications.**

| Algorithm | Domain | Strengths | Limitations | Main performance and representative cases |
|---|---|---|---|---|
| MHMPA | Cloud task scheduling | Efficient computation; Pareto-optimal solutions | Static model; lacks adaptability | 29.1% makespan reduction; energy and carbon savings (*Abdel-Basset et al., 2021*) |
| ML-MOO | Adaptive dynamic scheduling | Adjusts weights in real time; high adaptability | High computational cost | Responsive scheduling under dynamic demand (*Pinki Kumar et al., 2025*) |
| MIP | CHP system optimization | Accurate for constrained settings | Sensitive to demand variation; complex computation | 7.5% system efficiency gain; 15% boost in self-consumption (*Algieri, Morrone & Bova, 2020*) |
| ML prediction model | Renewable energy scheduling | Nonlinear modeling; highly adaptive | Requires high-quality data; weak on rare events | 87.5% simulation time cut; faster urban assessment (*Li et al., 2022b*) |
| Gradient-based optimization | Logistics distribution | Fast convergence; suitable for local optima | Prone to local minima; poor global exploration | 12% transport cost cut; 9.5% carbon emission drop (*Jiao & Zhang, 2025*) |
| WSM | Cost-load-renewable trade-off | Computationally simple; handles structured trade-offs | Highly weight-sensitive | Total system cost up 15.1%; industrial +21.4%, residential +27.3% (*Stoyanova & Monti, 2019*; *Zhao et al., 2023*) |

an Agile GSA (AGSA) variant improved 5G vehicular network efficiency by 57.43% over GA and PSO. Although GSA exhibits strong performance in network and mobility optimization, its application in urban energy and thermal management remains underexplored.

As smart cities continue to demand real-time, high-efficiency optimization, hybrid GSA–AI models represent a promising direction. The transition from pure physics-based techniques to hybrid metaheuristic–physics approaches reflects an emerging trend in urban optimization, where algorithms like SA and GSA are becoming essential for balancing computational efficiency with solution precision.

Table 11 summarizes representative physics-based optimization algorithms applied in smart city contexts, highlighting their domains, strengths, limitations, and real-world performance outcomes.

**Machine learning-enhanced optimization.** Machine learning-enhanced multi-objective optimization (ML-MOO) has demonstrated significant potential in a wide range of smart city applications. Compared to traditional optimization methods, ML-MOO combines machine learning's predictive capabilities with the decision-making power of MOO, resulting in substantial gains in computational efficiency and reduced reliance on costly simulation models. Applications span energy management, transportation optimization, environmental monitoring, and infrastructure planning. However, ML-MOO also faces persistent challenges, including limited interpretability, constrained generalization, and high computational demands.

ML-MOO has notably improved computational efficiency. For example, *Liu et al. (2024a)* employed an XGBoost + NSGA-II framework for urban energy consumption optimization, achieving a 420× acceleration compared to traditional CFD simulations. Similarly, *Shafiq et al. (2024)* utilized a CNN–LSTM model for thermal regulation in smart

**Table 11 Physics-based optimization algorithms for smart city applications.**

| Algorithm | Domain | Strengths | Limitations | Main performance and representative cases |
|---|---|---|---|---|
| CFD | Urban heat transfer, energy-efficient infrastructure | High physical accuracy; effective thermal modeling | Computationally expensive; not real-time applicable | Improved heat dissipation; reduced noise (*Cao et al., 2024*) |
| CNN-LSTM | HVAC, smart buildings | Learns temporal patterns; dynamic adaptability | No physical interpretability; training data dependency | 15.7–22.3% energy gain; 21.8–28.5% comfort boost (*Shafiq et al., 2024*) |
| Pareto MOO | Heating-electricity-water integration | Balances cross-domain objectives; boosts sustainability | High computational cost; not suited for real-time | 12–18% rise in urban energy efficiency (*Stoyanova & Monti, 2019*) |
| SA | Waste, logistics routing | Strong routing performance; good convergence in logistics | Narrow scope; limited to non-energy applications | Improved route efficiency and waste collection (*Salehi-Amiri et al., 2022*) |
| GSA | 5G vehicular networks, cyber-physical systems | Reduces path-loss; enhances network performance | Mostly network-oriented; underused in energy domains | 57.4% better path-loss *vs*. GA, PSO (*Okafor et al., 2024*) |

buildings, reducing energy usage by 22.3% while improving thermal comfort by 28.5%. In transportation, *Dong et al. (2024)* adopted deep reinforcement learning (DRL) for adaptive traffic signal control, which substantially reduced vehicle wait times and enhanced flow efficiency.

Despite these advancements, interpretability remains a core limitation. *Yuan et al. (2024)* applied SHAP analysis to enhance the interpretability of XGBoost in flood risk optimization, yet the method still lacked complete transparency in decision-making. Similarly, *Jalali et al. (2024)* achieved a 47% reduction in prediction error for air pollutant forecasting using an HNN + COOT model, but the opaque contribution of individual predictors hindered its usability in policy settings.

Generalization also remains a critical bottleneck. *Wu et al. (2024)* applied a Pix2Pix GAN + Genetic Algorithm approach for Beijing Hutong renovation optimization, but the model's dependence on high-quality training data limited its adaptability to shifting data distributions. In the domain of water resource management, *Tan & Yao (2023)* found that ML-based optimization models failed to generalize effectively across different geographic regions, highlighting the challenge of transferability.

In addition to interpretability and generalization concerns, high computational cost remains a practical barrier. *Preuveneers, Tsingenopoulos & Joosen (2020)* reported that deep learning-based optimization increased computational overhead by over 30% compared to conventional algorithms in IoT and edge computing settings, thereby restricting deployment feasibility. Moreover, the majority of ML-MOO applications are currently confined to data-rich and infrastructure-abundant environments. Table 12 summarizes representative ML-enhanced optimization algorithms, highlighting their strengths, limitations, and empirical performance across diverse smart city domains.

To address these limitations, future research must prioritize (1) enhancing model transparency through symbolic regression and causal inference, (2) improving generalization using transfer learning and self-supervised learning to reduce reliance on

**Table 12  ML-enhanced optimization algorithms for smart city applications.**

| Algorithm | Application domain | Strengths | Limitations | Main performance and representative cases |
|---|---|---|---|---|
| XGBoost + NSGA-II | Urban energy optimization | High efficiency; substitutes CFD models | Limited interpretability; sensitive to data quality | 420× faster than CFD (*Liu et al., 2024a*) |
| CNN-LSTM | Smart building HVAC | Handles temporal patterns; enhances energy use | High training cost; generalization challenges | 22.3% energy savings; 28.5% comfort gain (*Shafiq et al., 2024*) |
| DRL | Traffic signal control | Adaptive; supports real-time decisions | Data-intensive; costly training | Reduced vehicle wait times; improved flow (*Dong et al., 2024*) |
| HNN + COOT | Air pollution forecasting | Accurate; supports multi-objective learning | Difficult to interpret; low transparency | 47% MAE reduction ($NO_2$, $SO_2$) (*Jalali et al., 2024*) |
| Pix2Pix GAN + GA | Urban thermal design | Adapts to complex layouts; robust design support | Needs large, clean datasets | Improved ventilation and thermal comfort (*Wu et al., 2024*) |

**Table 13  Guidelines for selecting optimization algorithms based on problem characteristics and smart city application scenarios.**

| Optimization requirement | Algorithm category | Recommended algorithms | Application scenarios |
|---|---|---|---|
| High-dimensional MOO | Bio-inspired optimization | NSGA-III, MOGA | Urban land-use planning, large-scale energy optimization |
| Multi-objective trade-offs in resource management | Mathematical optimization | MOEA/D, weighted sum method (WSM) | Water distribution, logistics optimization |
| Computationally efficient surrogate modeling | ML-enhanced optimization | XGBoost + NSGA-II | Urban energy simulation, renewable energy scheduling |
| Time-sensitive dynamic scheduling | ML-enhanced optimization | DRL, PSO | Adaptive traffic signal control, IoT sensor task allocation |
| Improved solution diversity in optimization | Bio-inspired optimization | NSGA-II, EMOCA | Smart city network resilience, energy demand forecasting |
| Handling uncertainty and rare events | ML-enhanced optimization | HNN + COOT, Pix2Pix GAN + GA | Air pollution forecasting, urban building retrofitting |
| Multi-criteria decision-making | Mathematical optimization | MIP, Gradient-based optimization | Urban logistics land allocation, energy efficiency planning |
| Physics-based modeling with high accuracy | Physics-based optimization | CFD, SA | Heat transfer modeling, district heating network optimization |
| Real-time decision making under constraints | Hybrid ML + Metaheuristics | Adaptive MOGA, Hybrid PSO | Urban traffic optimization, farmland conservation |
| Computational cost-sensitive tasks | Mathematical optimization | MHMPA, WSM | Cloud computing task scheduling, cost-effective energy dispatch |

labeled data, and (3) optimizing computational efficiency through lightweight architectures, model quantization, and edge AI deployment. By tackling these challenges, ML-MOO can evolve into a more interpretable, adaptable, and practical framework for managing complex, data-intensive smart city systems.

Based on the characteristics of different optimization problems, Table 13 provides general guidelines for selecting the most suitable algorithm depending on the problem structure and application scenario.

**Table 14 Research gaps in MOO implementation.**

| Research gap | Challenges and opportunities | Key references |
|---|---|---|
| Computational efficiency | Real-time algorithms for large-scale data processing are still lacking. | *Wang (2020)*, *Wang et al. (2023, 2025)*, *Abdel-Basset et al. (2021)*, *Zhao et al. (2021)*, *Alharbi et al. (2022)*, *Yaman, Van Der Lee & Iacca (2023)*, *Dong et al. (2024)*, *Wu et al. (2024)* |
| Adaptive frameworks | Need for cross-domain adaptive frameworks in heterogeneous urban contexts. | *Wicki et al. (2021)*, *Wang et al. (2023)*, *Xiao (2024)*, *Zhang et al. (2024b)*, *Alghamdi (2024)* |
| Dimensionality reduction | High-dimensional MOO remains under-addressed in terms of decomposition and visualization. | *Zhang et al. (2024b)*, *Wu et al. (2024)*, *Yuan et al. (2024)*, *Wang et al. (2025)* |
| Uncertainty management | Limited adoption of robust/probabilistic MOO in stochastic urban systems. | *Yu et al. (2020)*, *Wang et al. (2021a)*, *Wang, Han & De Vries (2022)*, *Wu et al. (2024)*, *Zhao, Li & Wang (2025)* |
| Ethical considerations | Data transparency, privacy, and fairness in decision-making pipelines are underexplored. | *Preuveneers, Tsingenopoulos & Joosen (2020)*, *Xie et al. (2022)* |
| Emerging technologies integration | Quantum, neuromorphic, and LLM-based hybrid methods lack scalable frameworks. | *Stoyanova & Monti (2019)*, *Zhou et al. (2020)*, *Yaman, Van Der Lee & Iacca (2023)*, *Reitberger et al. (2024)*, *Wu et al. (2024)*, *Xu et al. (2024)*, *Wang et al. (2025)* |
| Smart security management | Real-time privacy-preserving MOO frameworks for IoT applications are insufficient. | *Preuveneers, Tsingenopoulos & Joosen (2020)*, *Wang (2020)*, *Mpatziakas et al. (2022)*, *Yaman, Van Der Lee & Iacca (2023)*, *Zhao, Bian & Mei (2024)* |
| Sustainable urban development | The development of frameworks that balance urban growth with ecosystem sustainability is still insufficient. | *Zhou et al. (2019)*, *Yu et al. (2020)*, *Saeidi, Aghamohamadi-Bosjin & Rabbani (2021)*, *Wicki et al. (2021)*, *Natanian (2023)*, *Qu et al. (2024)*, *Yuan et al. (2024)*, *Panda, Muthuraman & Elsts (2024)* |
| Energy-environment trade-offs | Balancing energy goals with environmental resilience remains difficult. | *Zhou et al. (2020)*, *Wang, Han & De Vries (2022)*, *Alghamdi (2024)*, *Xu et al. (2024)*, *Yu et al. (2024)*, *Fan et al. (2024)*, *Zhao, Li & Wang (2025)* |
| Intelligent infrastructure | There is a need for more dynamic infrastructure management frameworks. | *Abdel-Basset et al. (2021)*, *Zhao et al. (2021)*, *Wang et al. (2021b, 2023)*, *Xiao (2024)*, *Zhang et al. (2024b)*, *Wu et al. (2024)* |
| Adaptive resource allocation | Dynamic resource allocation strategies need to be prioritized. | *Zhou et al. (2020)*, *Wang et al. (2023, 2025)*, *Cao et al. (2024)*, *Yu et al. (2024)* |
| Collaborative urban mobility | Multi-scenario mobility optimization is still fragmented. | *Wang et al. (2023, 2025)*, *Xiao (2024)*, *Zhao, Bian & Mei (2024)* |
| Public data governance | Urban data in public systems often face access restrictions, privacy regulations, and fragmented ownership, which hinders model training and deployment. | *Hu et al. (2022)*, *Fan & Cui (2021)*, *Kang et al. (2024)* |
| Multi-stakeholder trade-offs | Conflicting stakeholder objectives complicate Pareto interpretation, requiring contribution-aware models to improve fairness and transparency. | *Eskelinen & Miettinen (2012)*, *Garber, Sarkani & Mazzuchi (2017)*, *Sooktip & Wattanapongsakorn (2017)*, *Li, Lai & Yao (2023)*, *Takayanagi, Mizutani & Loucks (2011)* |
| Real-time decision constraints | Dynamic urban systems such as traffic control or emergency routing require rapid optimization under incomplete data and fluctuating conditions. | *Wang et al. (2025)*, *Rosas-Solórzano et al. (2023)*, *Liu et al. (2024b)* |
| Human-in-the-loop optimization | Purely algorithmic solutions often fail to incorporate expert and resident feedback, necessitating human-in-the-loop optimization for actionable outcomes. | *Ghosh et al. (2024)*, *Cajot (2018)*, *Meza et al. (2024)* |

# CHALLENGES AND GAPS IN CURRENT MOO RESEARCH

Despite significant advancements in MOO methodologies, several key challenges persist, particularly in the context of smart cities. Table 14 summarizes the key research gaps identified by the selected articles, highlighting areas requiring further research.

The research gaps identified above reflect a broad range of technical and practical limitations in applying MOO to smart city systems. Key challenges include computational inefficiencies and the lack of scalable, real-time algorithms capable of handling dynamic and data-intensive environments. Addressing these issues requires adaptive optimization techniques that can maintain performance across changing urban conditions.

Equally important is the development of generalized frameworks that support cross-domain adaptability across systems such as energy, mobility, and IoT. Given the high-dimensional and interconnected nature of urban data, dimensionality reduction and decomposition strategies are essential for improving convergence and interpretability.

Uncertainty modeling and ethical concerns further complicate MOO applications. Current models often overlook urban stochasticity and lack transparency in decision processes. Robust and probabilistic optimization methods are necessary to enhance both resilience and fairness. Meanwhile, integrating emerging technologies, such as quantum computing, neuromorphic hardware, and federated learning offers great promise, yet their incorporation into practical, scalable MOO frameworks remains limited.

Among these emerging technologies, large language models (LLMs) have recently gained attention for their ability to enhance urban optimization through semantic reasoning and context-aware understanding. As a novel class of AI-driven methods, LLMs can complement MOO by supporting prompt-guided search, adaptive response generation, and real-time decision support. Recent studies demonstrate their applicability in improving traffic forecasting through spatiotemporal attention (*Mahmud et al., 2025*), orchestrating cross-layer services in 6G-enabled digital twin networks (*Jiang et al., 2024*), and facilitating complex multi-agent decision-making (*Huang et al., 2024*). However, their integration poses unique challenges: the lack of prompt engineering strategies suited to Pareto reasoning, substantial computational overhead, inference latency, and limited explainability of trade-offs. Addressing these issues requires coordinated advances in hybrid MOO–LLM algorithm design, efficient prompt generation pipelines, and interpretable low-latency architectures tailored for urban systems.

In parallel, issues related to security and privacy remain pressing, particularly in IoT-enabled environments. Real-time, privacy-preserving MOO frameworks are essential to protect critical urban infrastructure. Likewise, balancing energy–environment trade-offs is central to sustainable development. Additionally, fragmented data governance structures and access restrictions continue to hinder the deployment of scalable MOO systems across urban agencies. Human-in-the-loop optimization is also underutilized, limiting the integration of expert and community feedback in practical deployments. Unresolved gaps in adaptive resource allocation and collaborative urban mobility further underscore the need for refined decision frameworks capable of handling multi-scenario, multi-stakeholder complexity.

Bridging these research gaps, including algorithmic scalability, uncertainty modeling, ethical alignment, and emerging AI integration will enable MOO to serve as a foundation for smart cities that are not only efficient and adaptive, but also transparent, secure, and accountable.

# FUTURE PERSPECTIVES AND NEEDS

## Framework for future evaluations

This guideline is synthesized from a comprehensive analysis of the selected studies in this systematic review. It provides a structured and reproducible framework to support future evaluations, particularly for researchers exploring MOO in smart city contexts. As illustrated in Fig. 9, the framework comprises six interrelated steps, each addressing a critical stage in the MOO process and elaborated in the following sections.

Clearly defining the problem is the foundation of any successful optimization effort. This involves identifying the specific urban system component under investigation (*e.g.*, transportation, energy, infrastructure), selecting relevant decision variables, formulating objective functions and constraints, and characterizing uncertainty parameters. Previous research on urban mobility (*Wang et al., 2025*), energy systems (*Yu et al., 2024*), and environmental planning (*Zhao, Bian & Mei, 2024*) underscores the importance of this stage. A well-formulated problem also necessitates thoughtful sampling strategies (*e.g.*, stratified sampling, bootstrapping) to generate a reliable dataset (*Wang et al., 2023*).

Constructing a representative database is essential for surrogate modeling and algorithm training. This step involves integrating large-scale urban data from IoT networks (*Zhao et al., 2021*) and multi-source datasets (*Wu et al., 2024*), while applying robust sampling methods that capture the temporal and spatial dynamics of smart city environments. Varying sample sizes and data generation strategies should be compared to assess their influence on modeling performance.

Assessing the relative impact of input variables through sensitivity analysis helps streamline the optimization process. Variables that exhibit limited influence on objectives can be held constant to reduce dimensionality and computational burden (*Wicki et al., 2021*; *Yuan et al., 2024*). When dealing with multiple conflicting objectives, computing global sensitivity indices across objectives aids in identifying the most influential variables (*Zhao, Bian & Mei, 2024*).

Developing accurate surrogate models requires a combination of preprocessing, algorithm selection, and validation. Data normalization and encoding of categorical variables are often necessary before partitioning the data into training and testing subsets. Validation techniques such as k-fold or leave-one-out cross-validation help ensure model generalizability (*Wu et al., 2024*). Selecting suitable machine learning algorithms and fine-tuning their hyperparameters, *via* methods like grid search or Bayesian optimization, can substantially improve model accuracy (*Yaman, Van Der Lee & Iacca, 2023*).

Optimizing multiple conflicting objectives involves selecting and configuring appropriate algorithms. Adaptive genetic algorithms have demonstrated success in energy-efficient routing (*Zhao, Bian & Mei, 2024*), while clustering-based genetic algorithms are effective in complex mobility scenarios (*Wang et al., 2023*). Comparative evaluations using metrics such as runtime, Pareto front cardinality, diversity, convergence, and hypervolume are recommended. Additional rounds of optimization under varying uncertainty levels can improve solution robustness (*Xu et al., 2024*).

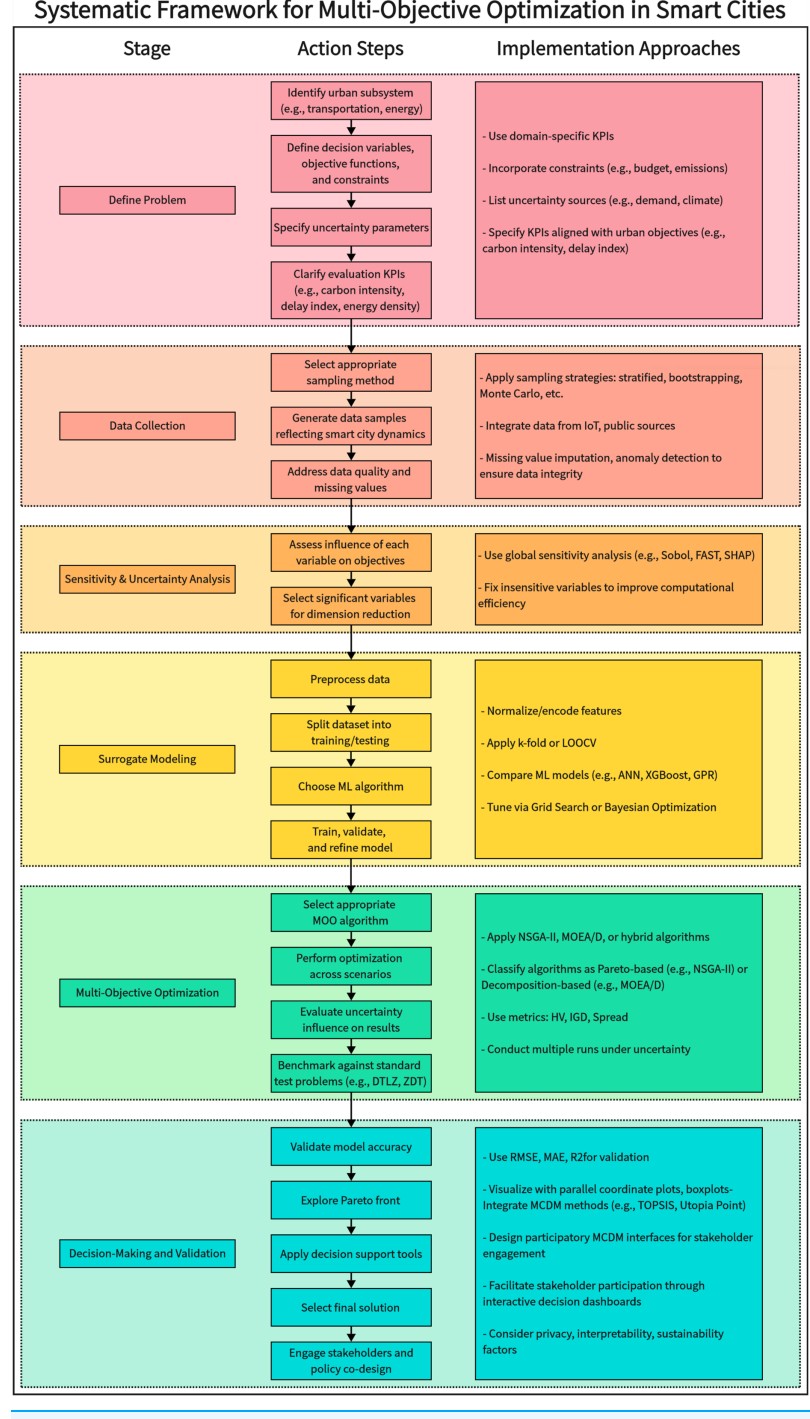

**Figure 9 Framework for future evaluations.**

Evaluating model validity and facilitating decision-making are critical for real-world deployment. Statistical performance indicators, such as $R^2$, mean absolute error (MAE), mean absolute percentage error (MAPE), and root mean square error (RMSE) should be used to assess surrogate accuracy. When model performance is inadequate, retraining with

expanded datasets or revised architectures may be needed. Multi-criteria decision-making (MCDM) techniques can support the selection of preferred solutions from the Pareto front. Visualization tools such as boxplots and parallel coordinate plots can further assist in exploring trade-offs and optimal value ranges (*Zhou et al., 2019*). Ethical considerations, such as data privacy (*Xie et al., 2022*) and environmental sustainability should be integrated into the decision process.

This framework offers a comprehensive guide for future evaluations, highlighting the importance of computational efficiency, adaptability, and ethical responsibility. By following this structured approach, future research can effectively address existing gaps and advance the development of intelligent, sustainable, and resilient smart cities through robust MOO strategies.

## Directions for future research in the field

Building on the findings and challenges identified in this review, several critical research directions are proposed to advance the integration of MOO in smart city development. These directions emphasize generalizability, computational efficiency, uncertainty management, and ethical integrity which are pillars essential for designing scalable and responsible urban optimization frameworks.

- Most surrogate models in current MOO studies are developed for specific buildings or single-use scenarios, limiting their applicability. Developing generalized surrogate models would require extensive multi-source datasets combined with data mining techniques, enabling broader applicability and improved predictive accuracy.
- Determining an optimal dataset size remains a key research area due to its impact on computational load, model accuracy, and optimization quality. Optimal sample size depends significantly on the selected machine learning algorithms and their effectiveness in capturing complex input-output relationships specific to each modeling scenario.
- Generating extensive datasets across diverse architectural scenarios is computationally intensive. Exploring transfer learning strategies, where models initially trained for specific tasks are adapted for related problems, can enhance efficiency and scalability by leveraging previously learned patterns, thus avoiding redundant computational effort.
- Sensitivity analysis should be systematically conducted as a prerequisite for optimization to evaluate the influence of design variables on objective outcomes. This process aids dimensionality reduction and narrows the optimization search space, thereby improving computational efficiency and model interpretability.
- Incorporating uncertainty parameters in surrogate model development and optimization processes is essential. Performing multiple optimization runs under varying extreme conditions of uncertainty can facilitate robust decision-making, allowing comparison across diverse optimal solutions and enhancing reliability.
- Developing intuitive, user-friendly surrogate-based optimization tools, such as software plugins or web-based applications is critical yet underexplored. These tools can significantly simplify building optimization processes, making advanced MOO methodologies accessible to both industry professionals and the broader community.

- Future studies should integrate a wider array of design variables beyond traditional envelope properties, carefully balancing conflicting objectives while considering real-world constraints, such as stakeholder interests and societal impacts. This holistic approach promotes practical, impactful optimization solutions in architectural design.

- Incorporating carbon emissions and environmental impacts explicitly as objective functions in optimization frameworks is crucial. Given the substantial role buildings play in global greenhouse gas emissions through energy use and material selection, prioritizing low-carbon design is integral to addressing climate change.

- MOO studies using surrogate models should be replicated across various climatic contexts to enhance the generalizability and robustness of sustainable building design solutions. Such cross-context validation will foster comprehensive knowledge and promote broader applicability of optimization strategies.

- Introducing and rigorously evaluating novel algorithms in machine learning and MOO is necessary to achieve performance improvements, enhance robustness, foster innovation, ensure methodological diversity, enable benchmarking, and improve interpretability of models.

- Researchers should consistently adopt and document ethical practices, including data and code transparency, explicit ethical disclosures, proactive identification of biases, rigorous model validation, and prioritizing interpretability. Such practices strengthen transparency, accountability, and trustworthiness in optimization research.

- Future research should also explore the integration of Large Language Models (LLMs) into MOO workflows to enhance semantic representation, constraint modeling, and human-in-the-loop decision-making in smart cities. LLMs can help extract implicit objectives, stakeholder preferences, and contextual trade-offs from unstructured data, thereby augmenting traditional optimization approaches with higher-level understanding and reasoning. Emphasis should be placed on designing lightweight, prompt-driven LLM-MOO architectures that can operate under edge or real-time conditions, while also developing structured templates and interpretability modules to ensure transparency, scalability, and consistent decision support in complex urban scenarios.

## CONCLUSION

This article presents a SLR focused on the application of MOO techniques within the context of smart city development. The review critically evaluates the integration of various optimization algorithms with emerging technologies such as machine learning, surrogate modeling, and digital twins. It highlights key opportunities and challenges by analyzing 117 articles covering diverse urban systems, including transportation, energy management, infrastructure development, and environmental sustainability.

The key findings from this SLR are as follows: Most reviewed studies originate from China, with urban systems related to transportation and energy receiving the most attention. Despite the potential of MOO in enhancing smart city resilience and sustainability, few studies explore cross-scenario adaptability, dynamic resource allocation,

and uncertainty management comprehensively. While Python and MATLAB are commonly used for modeling, there is limited comparison of machine learning algorithms, with artificial neural networks (ANN) and support vector machines (SVM) being the most prevalent. Hyperparameter optimization techniques are frequently overlooked; when utilized, genetic algorithms and grid search methods dominate. Sensitivity analysis is inconsistently applied, and the integration of uncertainty parameters in optimization processes remains underdeveloped.

Regarding optimization objectives, energy efficiency, cost reduction, and traffic management are prominent, while carbon emissions, social impact, and ecosystem services receive limited focus. NSGA-II emerges as the most widely adopted algorithm, with Pareto front exploration being common. However, few studies incorporate robust decision-making frameworks for selecting optimal solutions, which limits practical applicability. These findings are influenced by various factors, including dataset availability, computational complexity, and methodological convenience.

The review underscores that, although MOO applications in smart cities have gained significant traction, there remains substantial untapped potential. Determining ideal dataset sizes is essential to ensure desired accuracy levels in surrogate modeling and optimization outcomes. Integrating machine learning with traditional methods, such as sensitivity and uncertainty analyses, enhances robustness and computational efficiency, especially for complex urban systems. Additionally, variation ranges for each objective function must be evaluated, and adaptive weighting strategies should be applied based on specific urban contexts to avoid neglecting promising solutions.

Future research should prioritize the development of generalized MOO frameworks capable of addressing diverse urban scenarios. Large datasets representing various urban configurations are essential for creating robust and transferable models. To mitigate computational costs, data mining and transfer learning techniques should be employed. Furthermore, incorporating key objective functions such as social impacts, aesthetics, and climate change resilience is vital. The challenges associated with urban decarbonization emphasize the need to include carbon emissions as a central objective in optimization studies.

The creation of user-friendly tools based on surrogate models, compatible with urban planning software and accessible through web applications, would significantly promote the adoption of MOO approaches. Such tools could facilitate rapid assessments and optimizations, supporting stakeholders in developing sustainable urban solutions across different climatic contexts. This would foster broader application and validation of MOO strategies, advancing the resilience and adaptability of smart cities.

Finally, ethical considerations must be at the forefront of future research. Ensuring data transparency, making code openly available, and clearly stating ethical practices, such as addressing potential biases and ensuring model interpretability are crucial for maintaining research integrity. Promoting ethical standards will build trust within the research community and among practitioners, contributing to responsible and sustainable smart city development.

In summary, this review responds to three core research questions related to the evolution of MOO algorithms, their comparative performance across smart city domains,

and the identification of persistent challenges and future research directions. The findings reveal that integrating MOO with emerging technologies, such as machine learning, digital twins, and surrogate modeling offers a compelling pathway for achieving adaptive, efficient, and sustainable urban development. This synergy enhances simulation and decision-making capabilities while significantly reducing computational costs, enabling the delivery of technically robust and environmentally responsible smart city solutions. To fully realize this potential, researchers, policymakers, and practitioners must uphold a commitment to realistic, scalable, and ethically grounded methodologies that align technological innovation with societal needs.

## ACKNOWLEDGEMENTS

The authors acknowledge the use of ChatGPT (OpenAI) for language refinement and formatting assistance. All suggestions were reviewed and manually edited to ensure accuracy and alignment with the study's aims.

### Funding

This work was supported by Jiaxing Municipal Natural Science Foundation (No. 2025CGZ076). The funders had no role in study design, data collection and analysis, decision to publish, or preparation of the manuscript.

### Grant Disclosures

The following grant information was disclosed by the authors:
Jiaxing Municipal Natural Science Foundation: 2025CGZ076.

### Competing Interests

The authors declare that they have no competing interests.

### Author Contributions

- YiFan Chen conceived and designed the experiments, performed the experiments, analyzed the data, performed the computation work, prepared figures and/or tables, authored or reviewed drafts of the article, and approved the final draft.
- Weng Howe Chan conceived and designed the experiments, authored or reviewed drafts of the article, and approved the final draft.
- Eileen Lee Ming Su conceived and designed the experiments, authored or reviewed drafts of the article, and approved the final draft.
- Qi Diao conceived and designed the experiments, authored or reviewed drafts of the article, and approved the final draft.

### Data Availability

This is a literature review.

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
