# Peer review of "Multi-objective optimization for smart cities: a systematic review of algorithms, challenges, and future directions"

_PeerJ Computer Science, doi:10.7717/peerj-cs.3042_

## Round 0.1 · original submission · Major Revisions

Dear Authors,

Thank you for submitting your Literature Review article. Reviewers have now commented on your article and does not suggest to accept in its current form. We do encourage you to address the concerns and criticisms of the reviewers and resubmit your article once you have updated it accordingly.

Best wishes,

Reviewer 1 ·

Basic reporting

This paper should be rejected. Article has serious flaws, additional experiments needed, research not conducted correctly.
In my opinion scientific level of this publication is below requirements of this Journal. In particular, I do not see any breakthrough results compared to current and commonly known methods. Overall, the discussion and contribution of this paper is limited.
I would advise authors to deeply revise the paper to:
a) clearly explain their research questions,
b) to provide detailed description of the experiments and the results,
c) to ground the conclusions on facts coming from the experimentations.

Experimental design

.

Validity of the findings

.

Reviewer 2 ·

Basic reporting

Scope Clarity and Focus
While the paper claims to present a systematic review, the distinction between multi-objective optimization (MOO) in general and its specific application to smart cities is sometimes blurred. Consider tightening the scope by either:

Deepening the analysis on smart city domains (e.g., traffic management, waste optimization, energy efficiency), or

Adding a clearer taxonomy that categorizes MOO approaches according to smart city pillars.

Inclusion of Quantitative Comparisons
The review would benefit significantly from tables or figures comparing algorithms (e.g., NSGA-II, MOEA/D, SPEA2, etc.) across evaluation metrics (convergence, diversity, complexity) and application domains. This would improve the utility for researchers seeking practical guidance.

Experimental design

Missing Recent Trends in Large Language Models (LLMs)
With the rise of LLMs and foundation models being used for planning and optimization tasks, including a brief commentary or future direction on how generative AI or hybrid MOO-LLM frameworks might integrate into smart city applications would be valuable.

Evaluation Criteria Missing
There is a lack of discussion on benchmark functions, datasets, or performance indicators used in smart city MOO studies. Including a section that summarizes these would help standardize future research.

Depth of Challenge Discussion
The discussion on challenges is somewhat generic (e.g., scalability, data quality). Consider enriching this with domain-specific obstacles such as:

Validity of the findings

Data governance in public systems.

Multi-stakeholder trade-offs (e.g., between economic and environmental objectives).

Real-time decision-making constraints in dynamic urban systems.

---

## Round 0.2 · Minor Revisions

Dear Authors,

It is the considered opinion of one reviewer that the paper requires minor revisions. It is recommended that the concerns and criticisms raised by the reviewer be addressed and that the paper be resubmitted.

Best wishes,

Reviewer 3 ·

Basic reporting

The manuscript explores the important topic of multi-objective optimization (MOO) techniques and their impact on improving efficiency in smart city applications. The study is valuable and provides a comprehensive systematic review. Recommended improvements include enhancing clarity, consistency, and scientific rigor.

1. The term "Multi-Objective Optimization (MOO)" is first mentioned on line 53 and is subsequently referenced again on lines 276 and 284. Abbreviations must be defined upon their initial use and employed consistently thereafter. A comprehensive review of the manuscript is required to ensure consistency in all abbreviations.

2. The section labeled “Background and Motivation” within the Introduction fails to sufficiently address the motivational component of the study. The removal of this title is recommended. The section titled “Scope and Main Contribution” should be retitled as “Motivation and Main Contributions” to more accurately reflect the content and incorporate the rationale behind the study.

3. Contributions should be clearly itemized using bullet points or a numbered list to enhance readability and underscore the importance of each contribution.

4. The manuscript exhibits inconsistencies in the capitalization of proper nouns. For instance:

-In the first instance, "systematic literature review" is not capitalized, whereas in the second instance, it is capitalized. The inconsistency must be rectified across the text by adhering to a standard style guide, such as capitalizing only proper nouns or formal methodology titles as necessary

Experimental design

The manuscript explores the important topic of multi-objective optimization (MOO) techniques and their impact on improving efficiency in smart city applications. The study is valuable and provides a comprehensive systematic review. Recommended improvements include enhancing clarity, consistency, and scientific rigor.

1. The term "Multi-Objective Optimization (MOO)" is first mentioned on line 53 and is subsequently referenced again on lines 276 and 284. Abbreviations must be defined upon their initial use and employed consistently thereafter. A comprehensive review of the manuscript is required to ensure consistency in all abbreviations.

2. The section labeled “Background and Motivation” within the Introduction fails to sufficiently address the motivational component of the study. The removal of this title is recommended. The section titled “Scope and Main Contribution” should be retitled as “Motivation and Main Contributions” to more accurately reflect the content and incorporate the rationale behind the study.

3. Contributions should be clearly itemized using bullet points or a numbered list to enhance readability and underscore the importance of each contribution.

4. The manuscript exhibits inconsistencies in the capitalization of proper nouns. For instance:

-In the first instance, "systematic literature review" is not capitalized, whereas in the second instance, it is capitalized. The inconsistency must be rectified across the text by adhering to a standard style guide, such as capitalizing only proper nouns or formal methodology titles as necessary

Validity of the findings

The manuscript explores the important topic of multi-objective optimization (MOO) techniques and their impact on improving efficiency in smart city applications. The study is valuable and provides a comprehensive systematic review. Recommended improvements include enhancing clarity, consistency, and scientific rigor.

1. The term "Multi-Objective Optimization (MOO)" is first mentioned on line 53 and is subsequently referenced again on lines 276 and 284. Abbreviations must be defined upon their initial use and employed consistently thereafter. A comprehensive review of the manuscript is required to ensure consistency in all abbreviations.

2. The section labeled “Background and Motivation” within the Introduction fails to sufficiently address the motivational component of the study. The removal of this title is recommended. The section titled “Scope and Main Contribution” should be retitled as “Motivation and Main Contributions” to more accurately reflect the content and incorporate the rationale behind the study.

3. Contributions should be clearly itemized using bullet points or a numbered list to enhance readability and underscore the importance of each contribution.

4. The manuscript exhibits inconsistencies in the capitalization of proper nouns. For instance:

-In the first instance, "systematic literature review" is not capitalized, whereas in the second instance, it is capitalized. The inconsistency must be rectified across the text by adhering to a standard style guide, such as capitalizing only proper nouns or formal methodology titles as necessary

Additional comments

The manuscript explores the important topic of multi-objective optimization (MOO) techniques and their impact on improving efficiency in smart city applications. The study is valuable and provides a comprehensive systematic review. Recommended improvements include enhancing clarity, consistency, and scientific rigor.

1. The term "Multi-Objective Optimization (MOO)" is first mentioned on line 53 and is subsequently referenced again on lines 276 and 284. Abbreviations must be defined upon their initial use and employed consistently thereafter. A comprehensive review of the manuscript is required to ensure consistency in all abbreviations.

2. The section labeled “Background and Motivation” within the Introduction fails to sufficiently address the motivational component of the study. The removal of this title is recommended. The section titled “Scope and Main Contribution” should be retitled as “Motivation and Main Contributions” to more accurately reflect the content and incorporate the rationale behind the study.

3. Contributions should be clearly itemized using bullet points or a numbered list to enhance readability and underscore the importance of each contribution.

4. The manuscript exhibits inconsistencies in the capitalization of proper nouns. For instance:

-In the first instance, "systematic literature review" is not capitalized, whereas in the second instance, it is capitalized. The inconsistency must be rectified across the text by adhering to a standard style guide, such as capitalizing only proper nouns or formal methodology titles as necessary

---

## Round 0.3 · accepted · Accept

Dear Authors,

Thank you for addressing the reviewer's comments. The paper seems sufficiently improved and ready for publication.

Best wishes,

Reviewer 3 ·

Basic reporting

Dear Editor,
The authors' revisions have been carefully reviewed, and it appears that the article has been updated in accordance with my requested revisions. I accept the decision.

Experimental design

Dear Editor,
The authors' revisions have been carefully reviewed, and it appears that the article has been updated in accordance with my requested revisions. I accept the decision.

Validity of the findings

Dear Editor,
The authors' revisions have been carefully reviewed, and it appears that the article has been updated in accordance with my requested revisions. I accept the decision.

Additional comments

Dear Editor,
The authors' revisions have been carefully reviewed, and it appears that the article has been updated in accordance with my requested revisions. I accept the decision.